# Designing tailored combinations of structural units in polymer dielectrics for high-temperature capacitive energy storage

Rui Wang [1,2], Yujie Zhu[1,2], Jing Fu[1], Mingcong Yang[1], Zhaoyu Ran[1], Junluo Li[1], Manxi Li[1], Jun Hu [1], Jinliang He [1] & Qi Li [1] ✉

Many mainstream dielectric energy storage technologies in the emergent applications, such as renewable energy, electrified transportations and advanced propulsion systems, are usually required to operate under harsh-temperature conditions. However, excellent capacitive performance and thermal stability tend to be mutually exclusive in the current polymer dielectric materials and applications. Here, we report a strategy to tailor structural units for the design of high-temperature polymer dielectrics. A library of polyimide-derived polymers from diverse combinations of structural units are predicted, and 12 representative polymers are synthesized for direct experimental investigation. This study provides important insights into decisive structural factors necessary to achieve robust and stable dielectrics with high energy storage capabilities at elevated temperature. We also find that the high-temperature insulation performance would experience diminishing marginal utility as the bandgap increases beyond a critical point, which is strongly correlated to the dihedral angle between neighboring planes of conjugation in these polymers. By experimentally testing the optimized and predicted structures, an increased energy storage at temperatures up to 250 °C is observed. We discuss the possibility for this strategy to be generally applied to other polymer dielectrics to achieve further performance enhancement.

Dielectric capacitors are characteristic of ultrafast charging and discharging, establishing them as critically important energy storage elements in modern electronic devices and power systems. Polymer dielectrics have been the materials of choice for high-voltage dielectric capacitors by virtue of their high-voltage endurance and ease of processing[1,2]. However, these organic materials are confronted with huge challenges in the emergent applications such as mainstream technologies of renewable energy, electrified transportations and advanced propulsion systems, in which electronic devices are required to operate stably under harsh-temperature conditions ranging broadly from 150 °C to over 250 °C[3,4]. On the one hand, the commercially available capacitor films represented by biaxially oriented-

polypropylene (BOPP) are designed to be of wide bandgap ($E_g$) and large electrical resistance, but lack the thermal stability beyond 105 °C[5]. On the other hand, although the thermal stability of heat-resistant polymers such as polyimide (PI) and fluorene polyester (FPE) far exceeds (with the glass transition temperature ($T_g$) over 350 °C), they contain a rich number of highly conjugated structures, which cause exponentially increased conduction loss with increasing the temperature and electric field[6-8]. The construction of polymer-based composites has proved effective in promoting the high-temperature capacitive performance[9-11]. Nevertheless, the performance improvement in the composites also critically depends on the hosting polymers[12-15]. For example, the discharged energy density ($U_e$) of PI-

[1]State Key Laboratory of Power Systems, Department of Electrical Engineering, Tsinghua University, Beijing 100084, China. [2]These authors contributed equally: Rui Wang, Yujie Zhu. ✉e-mail: qili1020@tsinghua.edu.cn

based nanocomposite with the charge-discharge efficiency ($\eta$) > 90% is about 1 J/cm³ at 150 °C, significantly inferior to the room-temperature energy density (4 J/cm³) of the commercial benchmark capacitor film[16].

The high-electric-field capacitive performance and thermal stability tend to be mutually exclusive in the presently available polymer dielectrics[17–19]. This may arise from the fact that heat resistance is gained in these polymers via highly conjugated structures. Molecular modification to these polymers leads to appreciable $U_e$ (>2.0 J/cm³) only below 150 °C[20,21]. Very recently, a series of cyclic-olefin polymers with non-conjugated backbones and non-planar structures were synthesized, which aims at minimizing the impact of conjugation on $E_g$[22–24]. The resultant polymers with concurrently high $T_g$ (178–244 °C) and large $E_g$ (4.6–5 eV) show decent $U_e$ at 150 °C (5.7–8.7 J/cm³) and 200 °C (6.5 J/cm³). Yet, under the temperature extreme (at 200 °C), the high $U_e$ is achieved in these cyclic-olefin polymers at the expense of much reduced $\eta$ (~80% or below), which may cause massive waste heat production and thermal runaway of devices. Moreover, the cyclic-olefin polymers are not able to operate at 250 °C that is beyond the upper limit of $T_g$. Therefore, polymer dielectrics with excellent capacitive performance up to 250 °C remain unavailable, although they are urgently desired.

We noticed that, in many of the state-of-the-art high-temperature dielectric polymers and polymer composites, hopping conduction has been revealed to be the predominant mechanism of charge transport under high-temperature and high-electric-field conditions[9,10,20,22,25,26]. Different from some other conduction mechanisms, such as Ohmic or Poole-Frenkel emission, where electrons are mainly excited from the valence band or impurity levels to the conduction band, hopping conduction denotes the tunneling effect of electrons between adjacent charge traps in the material[27]. Such tunneling mechanism determining the possibility of electrons to escape is not directly associated

with the bandgap of materials, but rather is relevant to the local states (the depth of trap) associated with specific material structures. Therefore, we speculate that by merely focusing on the optimization of $E_g$ may not be able to yield the best outcome in the design of polymer dielectrics for high-temperature capacitive energy storage; Decisive structural factors in the polymer that are directly linked to the electrical performance at elevated temperatures should be figured out.

Herein, we report a tailored combination strategy for the design of polymer dielectrics towards extreme-temperature capacitive energy storage. By using a machine learning approach, we predict 110 types of PI-derived polymer structures from diverse combinations of 21 structural units that possess a wide distribution of electrical and thermal properties. We screen and synthesize 12 representative polymers all from commercial precursors in order to facilitate the large-scale production, and investigate their structures and performance systematically. By analyzing the experimental results in conjunction with computational simulations, we quantitatively determine the effect of each structural unit on the electrical and thermal properties of the resultant polymer, and unveil the key factor relevant to the capacitive performance at elevated temperatures. Based on the findings, we demonstrate that tailored combination of the structural unit can be implemented to construct the desired polymer dielectrics, leading to record energy storage performance at the temperature and electric field extremes.

## Results
### Machine learning for screening polymer structures
The development of polymer dielectrics for capacitive energy storage in extreme environments is usually driven by the trial-and-error method, accompanied by the shortcomings of high cost and low efficiency[20,28–30]. Machine learning is an emergent high-throughput

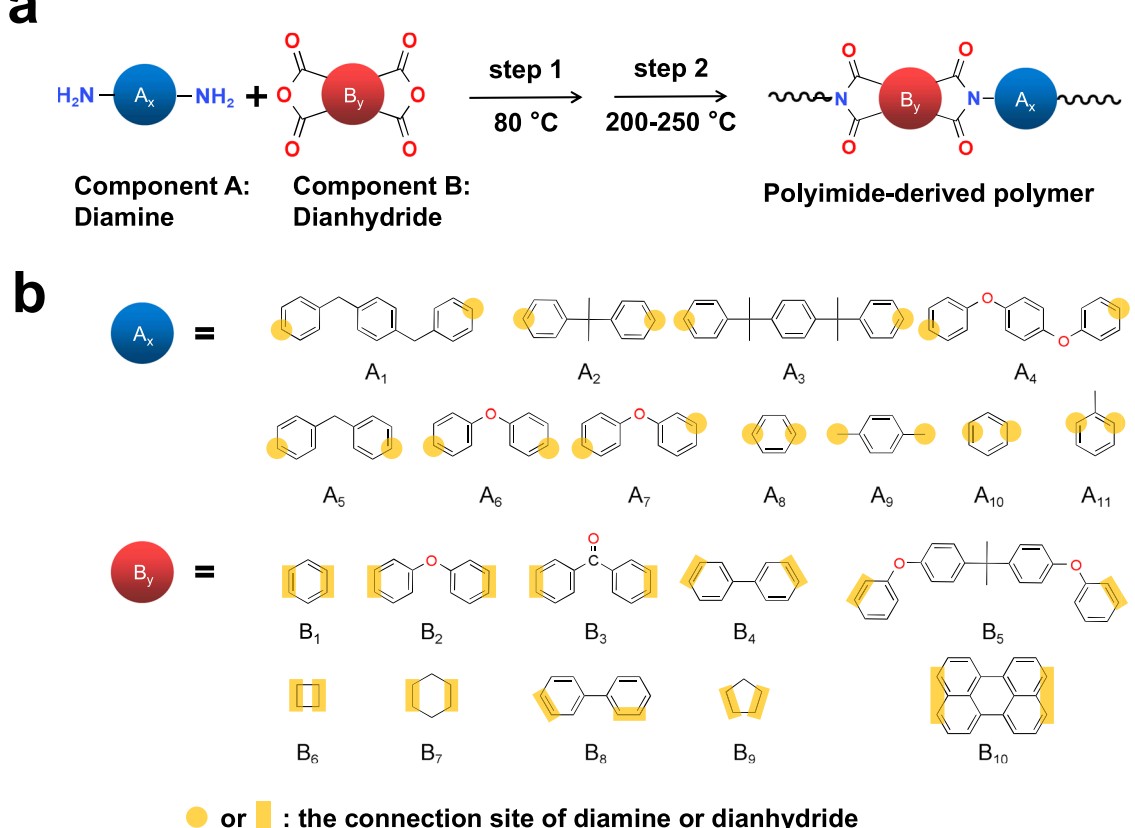

**Fig. 1 | Design of polyimide-derived high-temperature polymer dielectrics.**
**a** The schematic diagram for the synthesis route of polyimide-derived polymer dielectrics. These polymers are formed by condensation of the diamine monomer and dianhydride monomer. **b** The chemical structures of diamine monomers (A1–A11) and dianhydride monomers (B1–B10), which are represented by the blue ball and red ball, respectively.

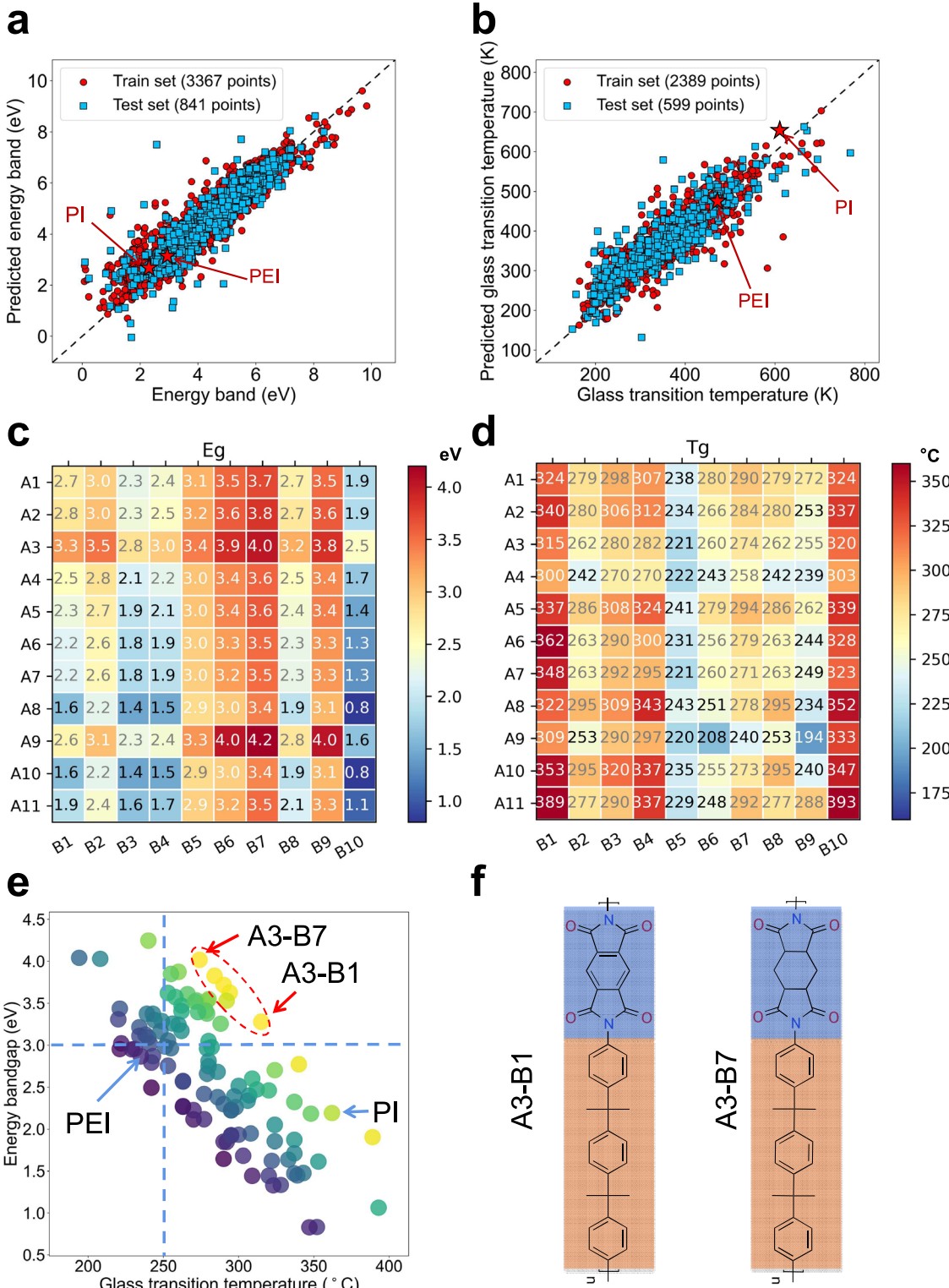

**Fig. 2 | Prediction of electrical and thermal performance by machine learning.**
**a, b** The prediction results of the energy bandgap ($E_g$) and glass transition temperature ($T_g$) via GPR learning method. **c, d** The $E_g$ and $T_g$ from machine learning of the polymers composed by 11 × 10 monomers. **e** The scatter plot of $T_g$ versus $E_g$ of all the 110 possible PI-derived polymers. Two dashed lines corresponding to 3.0 eV and 250 °C are marked as guide of eyes. Coordinates for combinations of polymers are shown in Table S1 of the Supplementary Information. **f** The molecular structure of two selected PI-derived polymers with the most balanced $T_g$ and $E_g$.

technology to guide the experimental design and synthesis of materials[29–31], which is conducive to rapid construction of "material genomes". Here, we first utilize the machine learning approach to assist with the preliminary screening of the backbone structure of polymer dielectrics by focusing on the two attributes $E_g$ and $T_g$ of

materials, where $E_g$ is usually believed to be positively related to the insulation performance[32] and $T_g$ determines the upper limit of the temperature capability[24]. We take the PI family as the research platform, which are commonly used as heat-resistant polymers, and can be readily synthesized from the diamine and dianhydride monomers

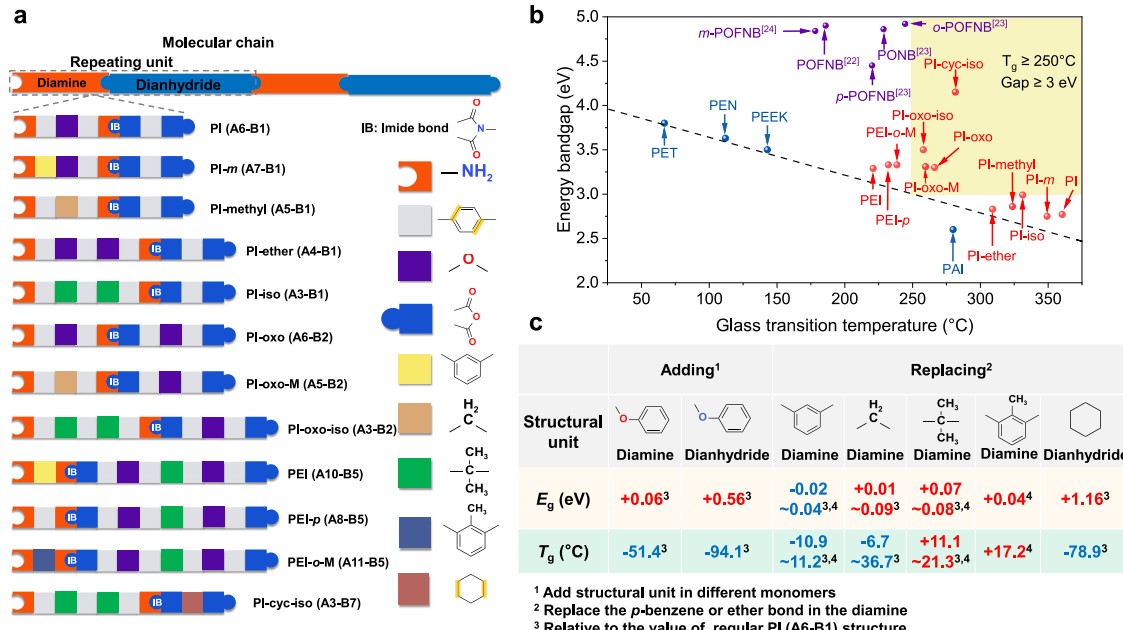

**Fig. 3 | Molecular structure and property characterization. a** Molecular structure puzzle of 12 polymers. We split the molecular structure into nine structural units, *i.e.*, amino, *p*-benzene, ether, anhydride, *m*-benzene, cyclohexane, *o*-toluene, methylene and isopropyl. **b** Comparison of electrical and thermal properties between polymers synthesized in this work, polymers reported previously and common commercial polymers (*i.e.*, polyethylene glycol terephthalate (PET), polyethylene naphthalate (PEN), poly(ether-ether-ketone) (PEEK), and polyamide-imide (PAI)). **c** The contribution of each structural unit on the glass transition temperature and bandgap relative to the regular PI or PEI structure.

through a two-step reaction (Fig. 1a and "Methods"). For the convenience of synthesis and large-scale preparation, some uncommon or hardly synthesized monomers are avoided in our selected chemical space of PI-derived polymers for prediction. We selected 21 typical structural units to compose 11 diamine monomers (A1–A11) and 10 dianhydride (B1–B10) monomers (Fig. 1b), and predicted the respective $T_g$ and $E_g$ of all the possible combinations (110 PI-derived polymers in total) using the gaussian processing regression (GPR) machine learning model (Supplementary Information for details). We selected 2988 polymer samples for the $T_g$ dataset and 4208 samples for $E_g$ dataset, in which the data points of classic PI and polyetherimide (PEI) polymers (Supplementary Fig. S1) were repeated in the dataset for serval times to improve the accuracy of the prediction (Fig. 2a, b). The $R^2$ parameters for $T_g$ test set and $E_g$ test set are 0.789 and 0.867, respectively, suggesting an accurate prediction for these two attributes.

The predicted $T_g$ and $E_g$ of all the 110 PI-derived polymers comprising one of the diamine monomers from A1 to A11 and one of the dianhydride monomers from B1 to B10 are shown in Fig. 2c, d, respectively, which can be further plotted in the form of $T_g$ versus $E_g$ (Fig. 2e). It is apparent that the two attributes ($T_g$ and $E_g$) are generally inversely correlated in the PI-derived polymers, implying that the constituent structural units (A1–A11 and B1–B10) may have distinct contributions to the two attributes. To learn the exact contributions of the structural units, we first selected the two PI-derived structures with the most balanced $T_g$ and $E_g$ among the 110 candidates as the target polymers (Fig. 2f), which are composed of A3 and B7, and A3 and B1 (marked with red circle in Fig. 2e), respectively. We also selected 10 more structures from the 110 candidates that resemble the molecular structures of the two target polymers for comparison (Supplementary Fig. S2).

### Understanding the critical role of structural units

All the 12 PI-derived polymers screened by machine learning were synthesized through the two-step reaction (Methods), and their molecular structures were verified by Fourier transform infrared (FTIR) spectroscopy and solid-state nuclear magnetic resonance (ssNMR) spectroscopy (Supplementary Figs. S3–S15). Each of the molecular structure in the 12 polymers can be seen as the combination of various structural units, as schematically illustrated in Fig. 3a. The naming of each PI-derived polymer depends on the modification to the molecular structure of regular PI (synthesized from A6 and B1) or PEI (synthesized from A10 and B5), *e.g.*, the PI-methyl refers to the structure formed by replacing the ether bond in the regular PI with a methylene bond. To investigate the influence of the constituent structural units on the electrical and thermal properties of the PI-derived polymers, the 12 polymers are divided into four groups for comparison. In the first group (Group 1), including PI, PI-*m*, PEI and PEI-*p*, we mainly compare the *meta*- and *para*-substituted benzene rings in the polymers. The second group (Group 2) containing PI, PI-methyl, PI-ether, and PI-iso is to reveal the difference between ether, methylene, and isopropyl bonds on the electrical and thermal properties of polymers. The third group (Group 3), including PI, PI-oxo, PI-oxo-M, and PI-oxo-iso, focuses on the analysis of flexible and rigid dianhydride units. The last group (Group 4) is to understand the impact of benzene ring modification on the properties of polymers, including side group-grafting on the benzene ring (PEI-*o*-M) or replacing it with a saturated six-membered ring (PI-cyc-iso).

The $E_g$ and $T_g$ of the 12 PI-derived polymers were experimentally determined by ultraviolet-visible (UV–Vis) absorption spectroscopy, and differential scanning calorimetry (DSC) and dynamic mechanical analysis (DMA), respectively (Supplementary Figs. S16–S18 and Table S2). Figure 3b compares the $E_g$ and $T_g$ of the polymers synthesized in this work with those of commercial polymers. The experimentally determined $E_g$ and $T_g$ are comparable to those predicted by machine learning (Supplementary Fig. S19), demonstrating the reliability of machine learning in preliminary screening of polymer structures. In the analysis of Group 1, we found that while the $T_g$ values of polymers with *p*-benzene structural unit are significantly larger compared to those containing the *m*-benzene structural unit (*e.g.*, 360.5 °C of PI

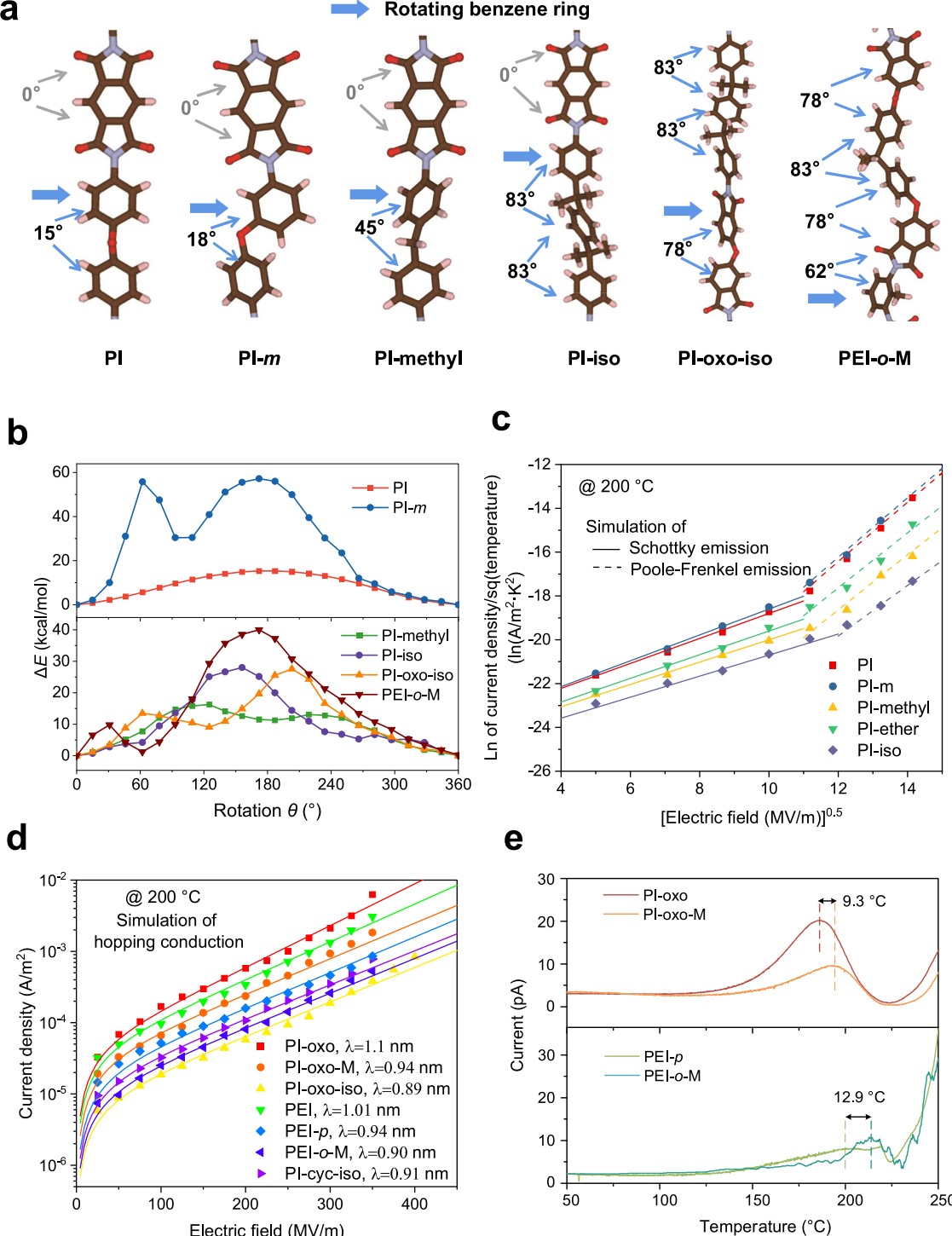

**Fig. 4 | Charge transport mechanism. a** Schematic diagram of six polymer chains at the lowest energy. **b** The rotational energy barrier of benzene ring as a function of rotation angle. **c** Current density of the PI-derived polymers synthesized from pyromellitic dianhydride as a function of electric field at 200 °C. **d** Current density of PI-derived polymers synthesized from long-chain dianhydride monomers as a function of electric field at 200 °C. **e** Thermally stimulated depolarization current of two groups of polymers with the same $E_g$.

versus 349.3 °C of PI-*m*, and 232.2 °C of PEI-*p* versus 221.3 °C of PEI), their $E_g$ values are quite similar (*e.g.*, 2.77 eV of PI versus 2.75 eV of PI-*m*, and 3.33 eV of PEI-*p* versus 3.29 eV of PEI). The characterization of Group 2 revealed that the isopropyl structural unit in the diamine monomer offers slightly improved $E_g$ and $T_g$ over the ether unit, *e.g.*, $E_g$ and $T_g$ of PI-iso increased by 0.16 eV and 22.2 °C, respectively, relative to PI-ether. By contrast, as suggested by the comparison in Group 3, the insertion of ether structural units into the dianhydride monomers

resulted in dramatic changes of $E_g$ (raised by 0.53 eV) and $T_g$ (declined by 94.1 °C). The comparison in Group 4 unveiled that the side group connected on the benzene ring is conducive to the concurrent promotion of $T_g$ and $E_g$ (raised by 17.2 °C and 0.04 eV, respectively), and that the substitution of benzene ring with cyclohexane structure results in significantly increased $E_g$ (raised by 1.16 eV) and decreased $T_g$ (declined by 49.7 °C). Based on the afore-mentioned experimental results, we are able to extract the exact impact of adding (or replacing)

each structural unit on the $E_g$ and $T_g$ values of the PI-derived the polymers, which are quantitatively benchmarked to the regular PI (*i.e.*, A6-B1) or PEI (*i.e.*, A10-B5) (Fig. 3c). For instance, the quantitative impact of adding an ether bond on the $E_g$ and $T_g$ values can be obtained by subtracting the data of PI-ether and PI. Similarly, by subtracting the data of PI-iso and that of PI-ether then dividing them by 2, we can get the quantitative impact of isopropyl relative to ether bond on the $E_g$ and $T_g$ values.

## Decisive structural factor to the high-temperature insulation performance

According to the prior notion[22–24], a larger $E_g$ of the polymer dielectric would result in greater high-temperature insulation and capacitive performance. Unexpectedly, we found that under high temperature (200 °C) and high-electric field (up to 400 MV/m) conditions, the measured leakage current density (Supplementary Fig. S25) of the PI-derived polymers (with the $T_g$ values all above 200 °C) does not comply with the trend of their respective $E_g$ values. For example, PI-cyc-iso with a high $E_g$ of 4.15 eV exhibits higher leakage current density (poorer insulation performance) than PI-oxo-iso with a lower $E_g$ of 3.5 eV. Therefore, we speculate that there are other factors affecting the high-temperature insulation performance of polymer dielectrics. We thus simulated the conjugation state of various PI-derived polymers in the lowest energy state through the molecular dynamics (MD) simulation (Fig. 4a), and determined the interaction between adjacent conjugated planes by comparing the rotational energy barrier obtained from rotating a ring structure around single bond (Fig. 4b). The impact of structural units on the interaction can be understood from two aspects. On the one hand, the conjugated planes in the molecular structure of PI and PI-*m* tend to be nearly coplanar (with the dihedral angle between adjacent benzene rings in the diamine <20°, Fig. 4a), and the rotational energy barrier with *p*-benzene structure (*i.e.*, regular PI) is significantly lower than that with *m*-benzene structure (16.3 kcal/mol versus 56.5 kcal/mol, Fig. 4b). In this case, the regular PI is less stable in the nearly coplanar state, and thereby is characteristic of weakened interaction with respect to the PI-*m*. On the other hand, the *ortho*-substituted methyl on the benzene ring, the ether bond in the dianhydride, and the isopropyl/methylene in the diamine all render the dihedral angle between adjacent benzene rings quite large (≥45°, Fig. 4a). In this case, the molecular structure with higher rotational energy barrier is more difficult to approach the coplanar state, giving rise to a stabilized state of weak interaction.

We found that the energy bandgap of the PI-derived polymers is not coupled with the dihedral angle between adjacent conjugated planes, as indicated by the density functional theory (DFT) computations of $E_g$ in the process that we progressively rotate one specific benzene ring in the polymer (Supplementary Fig. S27). To correlate the interaction between adjacent conjugated planes with the high-temperature insulation performance, we define the average dihedral angle ($\theta$) as the average measure of all the dihedral angles of adjacent conjugated planes in the diamine and dianhydride in one repeating unit of the polymer. For instance, in the PI-oxo-iso, there are three such dihedral angles with the measures of angle being 83°, 83° and 78°, and $\theta$ is calculated as (83° + 83° + 78°)/3 = 81.3°. Interestingly, it was unveiled that the trends of leakage current density (Supplementary Fig. S25) of the PI-derived polymers with $E_g$ above 3.3 eV match well with that of $\theta$ (Supplementary Tab. S3). The above analysis suggests that the interaction between adjacent conjugated rings is a decisive factor affecting the high-temperature insulation performance.

To understand why the interaction between adjacent conjugated rings plays a critical role, we investigated the conduction mechanisms of the materials. Since the leakage current of the material increases exponentially with the temperature, it is more meaningful to explore the conduction mechanism at higher temperature (*e.g.*, at 200 °C). It was revealed that, the pattern in which the leakage current density

measured under high temperature and high-electric field varies with the electric field is different between the polymers with $E_g$ above 3.3 eV and those with $E_g$ below 3.0 eV, clearly indicative of two distinct conduction mechanisms. To be specific, in the PI-derived polymers with rigid dianhydride (with $E_g$ < 3.0 eV), the fitting of the leakage current density versus electric field curve measured under 200 °C shows the transition from Schottky injection to Poole-Frenkel emission with the increase of electric field (Fig. 4c, and Supplementary Fig. S26a); By contrast, the fitting in the polymers with broken conjugated structure of dianhydride (with $E_g$ > 3.3 eV) conforms to the hopping conduction within the measured range of applied electric field (Fig. 4d, and Supplementary Fig. S26b). These results, along with the fundamental difference between Schottky injection/Poole-Frenkel emission and hopping conduction suggest that, in the case of polymers possessing relatively low $E_g$ (<3.0 eV), electrons can gain enough energy under high temperature and high-electric field to be injected from the electrode into the dielectric or to be excited from the valence band to the conduction band, whereas in the case of polymers with relatively high $E_g$ (>3.3 eV), they are more likely to be conducted through the tunneling effect between adjacent traps. Moreover, in the PI-derived polymers featuring hopping conduction ($E_g$ > 3.3 eV), the energy level of charge traps was found to be irrelevant to the $E_g$ value, as confirmed by the result of thermally stimulated depolarization current (TSDC, Fig. 4e) and calculated hopping distance (Supplementary Tab. S3). For example, while the depolarization current peak corresponding to the detrapping of charge carriers in PI-oxo is located at a lower temperature than that of PI-oxo-M (182.8 °C versus 192.1 °C, characteristic of a smaller trap depth in the former), these two polymers have exactly the same $E_g$ value ($E_g$ = 3.3 eV). This is also true in the comparison between PEI-*p* and PEI-*o*-M, whose depolarization current peaks are centered at 200.3 °C and 214.1 °C, respectively. Strikingly, we found that the trap depth in the PI-derived polymers is positively related to $\theta$ (Supplementary Tab. S3), meaning that a greater number of large dihedral angles dihedral angles render the tunneling of electrons more difficult. This is supported by the fact that the leakage current density shares the same trend to that of $\theta$. Summarizing all the results discussed in this section, a new insight can be gained, which challenges the conventional notion and might be critically important for the design of high-temperature polymer dielectrics for energy storage, *i.e.*, the high-temperature insulation performance may experience diminishing marginal utility as the bandgap increases beyond a critical point (about 3.3 eV in the PI-derived polymers) where the predominant conduction mechanism changes from Poole-Frenkel emission to hopping, and then the dihedral angle between adjacent plane of conjugation would become the decisive factor.

In addition, we evaluated the breakdown strength of PI-derived polymers at 150 and 200 °C, and found that the trend of leakage current is consistent with the breakdown strength (Supplementary Fig. S29), suggesting that electrical breakdown is the main mechanism causing the failure of PI-derived polymers. Subsequently, we characterized the dielectric properties of PI-derived polymers. Except that the dielectric constant of PI-cyc-iso decreases to 2.72 due to the replacement of benzene ring with cyclohexane, the dielectric constant of other polymers at high temperatures is similar (ranging from 3 to 3.4) owning to the fact that the structural unit all exhibit weak polarity (Supplementary Fig. S30). However, the dielectric dissipation shows frequency and temperature dependence, *e.g.*, the dielectric loss of PI-derived polymers with $E_g$ < 3.0 eV increases significantly at low frequencies ($10^2$–$10^3$), which can be attributed to the increase of the conduction loss. To further clarify the temperature stability of the dielectric properties of PI-oxo-iso, we evaluated the temperature-dependent dielectric spectrum of PI-oxo-iso ranging from 30 to 250 °C at 1000 Hz, and compared it with typical polymers (PI and PEI). As show in Supplementary Fig. S31, it was proved that the dielectric properties of PEI and PI-oxo-iso remain stable below the glass

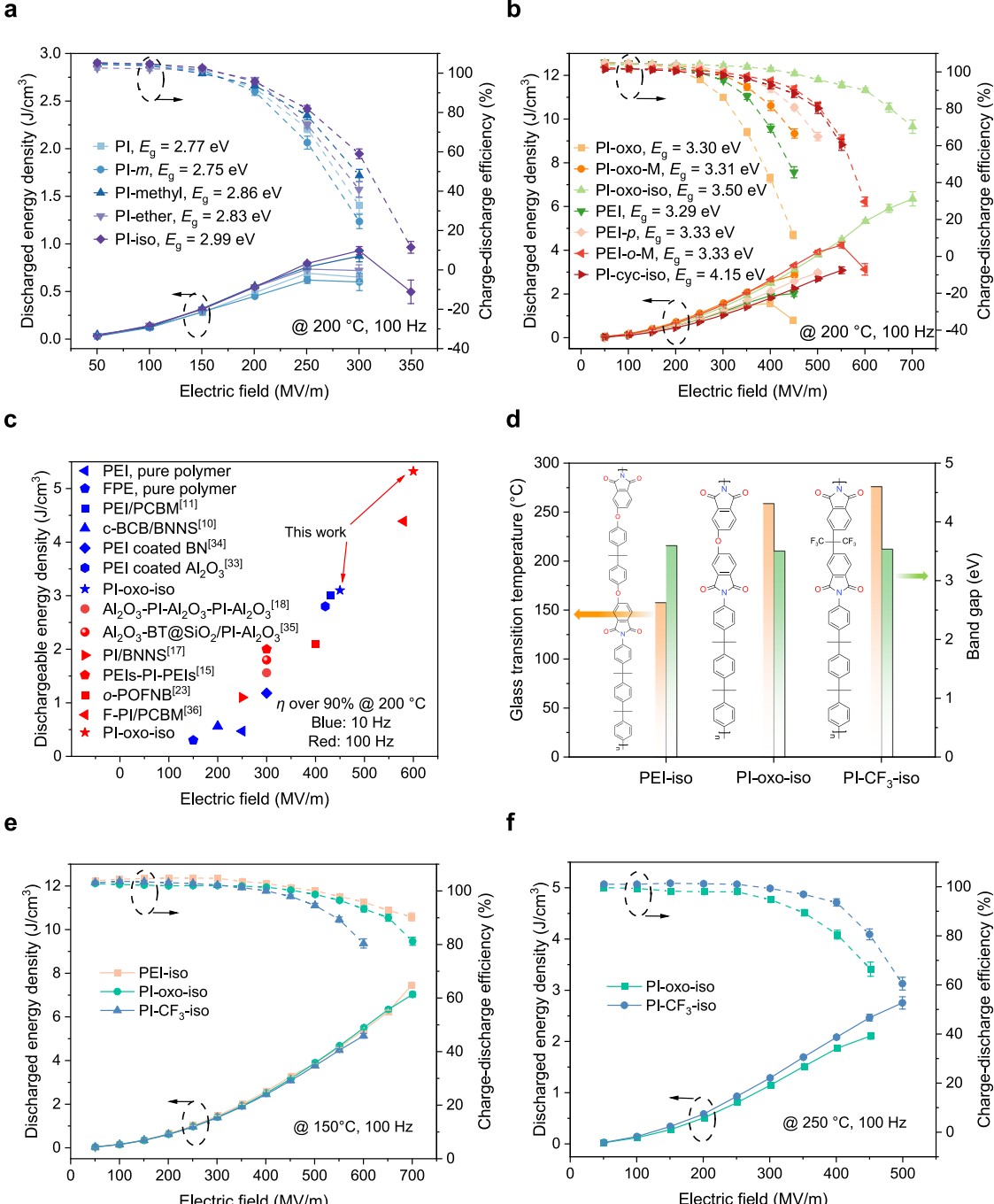

**Fig. 5 | High-temperature energy storage performance. a** High-temperature capacitive performance of the PI-derived polymers with $E_g$ below 3.0 eV at 200 °C. **b** High-temperature capacitive performance of the PI-derived polymers with $E_g$ above 3.3 eV at 200 °C. **c** Comparison of the maximum discharged energy density at above 90% efficiency between the PI-oxo-iso and the state-of-the-art polymer-based high-temperature dielectrics at 200 °C with 10 or 100 Hz applied electric field. **d** Comparison of glass transition temperature and bandgap of PEI-iso, PI-oxo-iso, and PI-CF$_3$-iso, the corresponding molecular structure formula is shown on the left side to the column diagram. **e, f** Discharged energy density and charge-discharge efficiency as a function of electric field of PEI-iso, PI-oxo-iso and PI-CF$_3$-iso at 150 °C and 250 °C. The average values and max–min error bars of the results were obtained from six parallel samples.

transition temperature, while PI presents a slightly decreased dielectric constant with the increase of temperature.

## Capacitive energy storage performance

To examine the new understandings, we next studied the energy storage performance of the PI-derived polymers by measuring the unipolar electric displacement-electric field (*D-E*) loops at high temperature (Fig. 5, and Supplementary Figs. S33−S44). It is apparent that the high-temperature energy storage performance ($U_e$ and $\eta$) of

the polymers with relatively low $E_g$ (<3.0 eV) complies with the trend of $E_g$. As $E_g$ increases from 2.75 to 2.99 eV, the $U_e$ and $\eta$ of polymers gradually increase from 0.6 J/cm³ and 25% to 0.9 J/cm³ and 59%, respectively, which can be attributed to the dependence of Schottky injection barrier and Poole-Frenkel emission activation energy on the $E_g$ (Fig. 5a). By sharp contrast, in the polymers with relatively high $E_g$ (>3.3 eV), in which hopping conduction is the predominant mechanism, the high-temperature energy storage performance is no longer in accordance with the pattern of $E_g$ (Fig. 5b). For example, while the $E_g$ of

PI-cyc-iso is higher than that of PI-oxo-iso (4.15 eV versus 3.5 eV), the $U_e$ of the former is much inferior to that of the latter at 150 °C (Supplementary Fig. S45) and 200 °C (Fig. 5b), which corroborates that by merely focusing on the optimization of $E_g$ may not be able to yield the best high-temperature capacitive performance. Furthermore, at 200 °C, the polymer with greater $\theta$ exhibits higher $U_e$, *e.g.*, PI-oxo-iso, with the largest $\theta$ (81.3°) of all the samples, possesses the highest $U_e$ (6.4 J/cm$^3$). This result solidifies the decisive role of the dihedral angle between adjacent conjugated planes in the design of PI-derived high-temperature dielectrics.

In addition to $U_e$, the maximum discharged energy density above 90% charge-discharge efficiency ($U_{e90}$) is even more important for the high-temperature energy storage[9,11]. This is because an energy density achieved at low efficiencies (~80% or lower) is inevitably accompanied with massive heat dissipation, and may cause thermal runaway of the device under high temperature that is close to the thermal limit of material. We compared the $U_{e90}$ of PI-oxo-iso with the previous best results reported in the literature (Fig. 5c, Supplementary Tab. S4). Impressively, the $U_{e90}$ of the polymer reported in this work is as high as 5.3 J/cm$^3$, surpassing all the existing pure polymers and their composites[10,11,15,17,18,23,33–36]. Note that, although the previous best high-temperature polymer dielectric (cyclic-olefin polymer)[23] delivers a high $U_e$ (6.5 J/cm$^3$) at 200 °C that is comparable to the PI-oxo-iso, its $U_{e90}$ is merely around 2 J/cm$^3$, suggesting that the PI-oxo-iso would significantly outperform the cyclic-olefin polymer at the temperature and electric field extremes. Besides, the PI-derived polymers reported in this work are all prepared with mature synthetic approach using commercial precursors, which facilitates large-scale production.

We further characterized the quality of the high-performance dielectric film (PI-oxo-iso) to evaluate its suitability for industrial production and application. The results of atomic force microscopy (AFM) and scanning electron microscopy (SEM) show that the PI-oxo-iso film is of a small surface roughness (Ra = 0.22 nm) without perceptible defect (Supplementary Figs. S47 and S48). The homogeneity of the film was investigated on a $20 \times 20$ cm$^2$ area by measuring the high-temperature capacitive performance, which is overall stable in different regions (Supplementary Fig. S49). The ability to form high-quality films was also confirmed in different film thicknesses with various electrode areas by testing the breakdown strength (Supplementary Figs. S50 and S51). In addition, the energy storage performance of the film exhibits decent cyclic and temperature stability (Supplementary Figs. S52 and S53), both of which are important for capacitor application.

With the specific function of each structural unit fully understood, we are able to customize polymer dielectrics with the optimal capacitive performance at different temperature conditions. The design concept is based on three primary principles: Firstly, $E_g$ should be higher than 3.3 eV to ensure the hopping conduction; Secondly, $T_g$ needs to be more than 10 °C higher than the target operating temperature to secure the thermal stability; Thirdly, as many as possible large angle conjugated rings should be added to the polymer chain to increase the trap depth. Based on these principles, we implemented tailored combination of the structural units, and synthesized 2 more polymers, *i.e.*, PEI-iso and PI-CF$_3$-iso designed for the applications at 150 °C and 250 °C, respectively (Fig. 5d, Supplementary Figs. S54–S56 for details). PEI-iso was obtained by adding ether and isopropyl structural units in the dianhydride of PI-oxo-iso, which inserted two large dihedral angles to the molecule and reduced the $T_g$ value by 81.5–91.7 °C relative to the PI-oxo-iso (therefore the $T_g$ of PEI-iso should be in the range of 166.5–176.7 °C). PI-CF$_3$-iso was designed by replacing the ether unit in the dianhydride of PI-oxo-iso with hexafluoroisopropylidene (similar to isopropyl, since the dianhydride containing hexafluoroisopropylidene is commercially available) to increase the $T_g$, and the $T_g$ of PI-CF$_3$-iso is predicted to be in the range of 269.3–279.5 °C. The experimental results show that the $E_g$ values of

PEI-iso and PI-CF$_3$-iso are 3.59 eV and 3.53 eV (Fig. 5d, Supplementary Fig. S58), respectively, and their $T_g$ values are 167.4 °C and 275.5 °C (Fig. 5d, Supplementary Figs. S59 and S60), respectively, exhibiting good agreement with the values predicted by the tailored combination. Outstandingly, PEI-iso shows the highest $U_{e90}$ (7.5 J/cm$^3$) at 150 °C among all the synthesized polymers in this study and other polymers and composites reported previously (Fig. 5e, Supplementary Fig. S63). More strikingly, PI-CF$_3$-iso possesses unprecedented energy storage performance at 250 °C ($U_{e90}$ of 2.1 J/cm$^3$) (Fig. 5f, Supplementary Figs. S64 and S65), which is over an order of magnitude larger than that of the best polymer and polymer composite ($U_{e90}$ less than 0.2 J/cm$^3$) reported in the literature[10]. The result from mechanical test shows that all the high-performance polymers including PI-oxo-iso, PEI-iso and PI-CF$_3$-iso have comparable mechanical strength to the regular PEI (Supplementary Figs. S23 and S57), which has been proved to be able to form thin capacitor films.

## Discussion

A tailored combination of structural units is demonstrated for the design of high-temperature polymer dielectrics. By using machine learning, polymer structures with a broad range of electrical and thermal properties are screened out, which enables grouped comparisons and quantitative description of the impact of each structural unit on the $E_g$ and $T_g$ of polymers. Besides, the prediction by machine learning also offers plentiful enough polymer structures with high $E_g$ and high $T_g$. In-depth experimental investigation on the electrical properties reveal that the high-temperature insulation performance would experience diminishing marginal utility as the bandgap increases beyond a critical point (about 3.3 eV) in these polymers, where the predominant conduction mechanism changes from Poole-Frenkel emission to hopping. MD and DFT simulations suggest that dihedral angles between adjacent conjugated planes in the molecule play a decisive role in determining the extreme-temperature capacitive performance. This is strikingly different from the prior notion that optimization of $E_g$ would yield the best capacitive performance at high temperature. Based on these understandings, three different polymer structures are designed via the tailored combination of structural units, for operation under 150, 200, and 250 °C, respectively, all exhibiting record discharged energy densities above 90% efficiency under the respective temperature conditions (7.5, 5.3, and 2.1 J/cm$^3$). The strategy reported in this work may offer new possibilities to resolve the challenging dilemma of achieving excellent capacitive performance at the temperature and electric field extremes.

## Methods

### Materials

Pyromellitic dianhydride, 4,4′-diaminodiphenyl ether, *m*-phenylenediamine, 3,4′-diaminodiphenyl ether, *p*-phenylenediamine, 5,5′-((propane-2,2-diylbis(4,1-phenylene))bis(oxy))bis(isobenzofuran-1,3-dione), and 4,4′-diaminodiphenylmethane were purchased from Sigma-Aldrich. α,α′-Bis(4-aminophenyl)−1,4-diisopropylbenzene, 4-bis(4-aminophenoxy)-benzene, 4,4′-(hexafluoroisopropylidene)diphthalic anhydride and 4,4′-oxydiphthalic anhydride were purchased from TCI. 2-Methyl-*m*-phenylenediamine and 1,2,4,5-cyclohexanetetracarboxylic dianhydride were provided by Anhui Zesheng Technology Co., Ltd.

### Synthesis of polymers

All polymers are synthesized by solution method. First, a certain mass of dianhydride monomer was added to 10 mL of N-methyl-2-pyrrolidinone (NMP) solution. Then, an equal molar amount of diamine monomer was added to the mixed solution, followed by stirring at 80 °C for 12 h to ensure that the diamine and the dianhydride monomer reacted to form a poly(amic acid). Due to the different molecular weights of dianhydride and diamine monomers, the total

monomer mass in the various solutions we prepared here was always kept within the range of 300–400 mg, and the specific amount of monomers in all polymer synthesis processes is shown in Table S5 (Supplementary Information). Afterwards the reacted solution was cast on a pre-cleaned glass plate and dried at 80 °C for 12 h. To make the poly(amic acid) undergo the imidization reaction, we need to conduct additional high-temperature treatment on the sample. The polymers were heat-treated in an oven by a gradient heating method, kept at 100, 150, 200, and 250 °C in sequence for 1 h. After naturally cooling to room temperature, the film was peeled off from the glass substrate in water, and vacuum-dried in a 100 °C oven to remove residual moisture.

## Characterization

Solid-state NMR carbon spectra were recorded on a JNM-ECZ600R spectrometer. In the 13C NMR test, the resonance frequency was 150 MHz, the relaxation delay was 5 s, the linewidth factor was 80 Hz, the mas frequency was 12 kHz, and the contact time was 2 ms. Fourier transform infrared spectra (FTIR) were recorded on a Thermo Scientific Nicolet iS10 spectrometer using the attenuated total reflectance (ATR) mode. Differential scanning calorimetry (DSC) was carried out by a DSC system (TA Instrument Q100, USA) from 30 °C to 400 °C at 10 °C/min. Dynamic mechanical analysis (DMA) was performed by a TA Instruments DMAQ800 at a heating rate of 5 °C/min. UV–vis absorption spectra were measured by a Hitachi U-3010 spectrophotometer ranging from 200 to 700 nm with wavelength accuracy of ±0.3 nm. Thermogravimetric analysis (TGA) was performed by a Thermogravimetry (TA Instruments Q5000IR, USA) ranging from 30 °C to 800 °C at 10 °C/min under a nitrogen atmosphere. Atomic force microscopy (AFM) for surface morphology were obtained by a Bruker Dimension Icon atomic force microscope in tapping mode. Gel permeation chromatography (GPC) was used on a high-performance liquid chromatography (Waters E2695 Alliance) to obtain molecular weights, where the solvent was DMA. In the mechanical property test, the sample was cut into a long strip of 10 mm wide and 40 mm long (the effective length of all samples is 30 mm), and stretched at a rate of 2 mm/min until it breaks to obtain the stress-strain curve. The slope of the tangent under 5% strain is defined as Young's modulus. Scanning electron microscopy (SEM) were conducted on a ZEISS MERLIN Compact field emission electron microscope. Before measuring the electrical and dielectric properties, aluminum electrodes were evaporated on both sides of the film using a BENCH TOP TURBO evaporative coating instrument from Denton Vacuum. A Novocontrol Concept 80 dielectric spectroscopy meter equipped with a Quatro-Cryosystem temperature control system was used for obtaining dielectric constant and dissipation factor, the electrode diameter used in this test was 10 mm. Leakage currents were performed on a Keithley 6517B amperemeter, and the voltage and temperature were controlled by a Stanford PS350 voltage source and an oven, respectively. Dielectric breakdown strength was recorded on a TREK 610C amplifier at a voltage rise rate of 500 V/s, 15 data were obtained for each sample and analyzed by the two-parameter Weibull statistic. In the dielectric breakdown strength and leakage current tests, the electrode diameter of the sample was 2 mm. The electric displacement–electric field (D–E) loops were recorded on a modified Sawyer-Tower circuit under a unipolar wave at the frequency of 10/100 Hz, and the electrode diameter used in this test was 3 mm. In the breakdown strength and D–E loops tests, the testing samples were soaked in dimethylsiloxane, and the temperature of silicone oil was controlled by a hot plate equipped with a temperature sensor. The thermally stimulated discharge current (TSDC) tests were measured by a Keithley 6517B amperemeter equipped with a Quatro-Cryosystem temperature control system. First, all samples were polarized under 50 MV/m DC poling electric field for 30 min at 200 °C, then quickly cooled to 30 °C with the electric field remaining 50 MV/m. Finally, removed the electric field and short-

circuit the sample after holding at 30 °C for 10 min, heated the sample to 300 °C at a heating rate of 3 °C/min, and recorded the current at each temperature point.

## Machine learning

The $T_g$ and $E_g$ prediction model for PI-derived system is trained by polymers from linear polymer chemical space, which is collected from PolyInfo dataset[37] and Kahazana dataset[38]. Particularly, Due to fact that the calculated $E_g$ values are usually lower than the experimental ones, we replaced some $E_g$ values of PI-derived polymers in the training dataset with experimental data from previous literature[11,39] to improve the accuracy of prediction. We also took the K-fold ($k = 5$) method, where all the data except for those of the PI-derived polymers in the training dataset are divided into five parts and the data of the PI-derived polymers are trained in every part. The dataset has been plotted in Fig. S1 where the PI-derived polymers are highlighted. The $T_g$ dataset contains 2988 samples collected form PolyInfo dataset, and the $E_g$ dataset contains 4208 samples collected form Kahazana dataset. The datasets contain 7 elements, i.e., C, H, O, N, S, F, Cl. The fingerprint contains two parts.

One is collected from the traditional molecular fingerprint, which represents the content of fragment inside the monomer unit. The fragment is collected from MACCS public keys and the basic groups form dataset polymers. Moreover, different molecular fingerprint features are also added, like BalabanJ, TPSA, etc., which can be achieved by the RDKit python package. The other part contains the unique feature of polymer monomer unit, compared with the molecular fingerprint, like the length of the main chain. Overall, 781 dimension fingerprint has been generated for prediction. The machine learning method for prediction is the Gaussian processing regression (GPR), where rational quadratic (RQ) kernel is introduced for selection combined with the fingerprint engineering method (RFE). Considering that the minor difference between the predicted results and the experimental results is acceptable (Fig. S19), the machine learning method Gaussian processing regression (GPR) model is applied rather than other machine learning methods (neural network, random forest, decision tree, etc.). In the prediction, 80% of the dataset are divided into the training set and 20% are set to the test set. In the training set, k-fold cross validation ($k = 5$) are introduced for increasing the training set scale.

## Density functional theory methods

The density functional theory (DFT) has been performed in the software VASP, where the self-consistence calculation tends to underestimate bandgap $E_g$ of dielectric polymer by 30% or more[40–42]. Thus, Heyd–Scuseria–Ernzerhof (HSE06) hybrid function is employed in the bandgap calculation of the single PI polymer chain[43]. However, the bandgap values of polymers are still underestimated. The results are only used for qualitative analysis here. In addition, the rotation energy is also calculated via this method.

## Data availability

The data that support the findings of this study are available from the authors on request.

## Code availability

The code that supports the findings of this study is available from the authors on request.

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

## Acknowledgements

This work was supported by the National Natural Science Foundation of China grants 51922056 (Q.L.), 92166203 (Q.L.) and 51921005 (J.L.H.).

## Author contributions

Q.L. conceived the idea. Q.L. and R.W. designed the experiments. R.W., J.F., M.C.Y., Z.Y.R. and J.L.L. carried out the experiments. Y.J.Z. and M.X.L. performed the simulations. Q.L., J.L.H., J.H. and R.W. analyzed the data. Q.L. and R.W. wrote the manuscript. All authors discussed the results and commented on the manuscript.

## Competing interests

The authors declare no competing interests.
