## [Peer Review File · Nature Communications]

Designing tailored combinations of structural units in polymer dielectrics for high-temperature capacitive energy storageREVIEWER COMMENTS

Reviewer #1 (Remarks to the Author):

The authors reported a rational approach to understand and design polyimide polymers for high temperature dielectrics. Based on the machine-learning results, PIs with decent E_g s and T_g s were identified and synthesized. Experimental studies indicated that the conduction mechanism altered from Poole-Frenkel emission to hopping once above an empirical threshold (3.2 eV), and in the hopping region, the dihedral angle between adjacent conjugated ring systems becomes decisive to the trap depth and hence the leakage current density. The findings enabled the design and synthesis of new PI polymers with impressive high-T energy storage properties. Both the approach and the materials properties are state-of-the-art. Despite some technical inconsistencies, the work is quite commendable for the systematic approach, and the main findings could be inspirational for designing aromatic main chain dielectric polymers. I thus recommend its acceptance after the authors address the following comments:

Main concerns:

1. In Figure 5b, the withstand voltage of PI-oxo-iso material is very high, 700 MV/m. However, in Supplementary Fig. S22, the breakdown strength of this material at the same temperature is only 606 MV/m, which is much lower than 700 MV/m. The inconsistency in the results of this experiment deserves further scrutiny. In other words, can each sample reach 650 or 700 MV/m in capacitor tests? This raises the concern about the lack of error analysis, as error bars are missing from the major performance plots presented in Figure 5. Since this work reports many numerical values, it is critical to know if the results are average quantities or best-case values. Although the authors state that 15 specimens were tested for each polymer in the electric breakdown measurements, it is not obvious whether the reported values for other characterizations are representative.
2. Page 7, the first sentence of the second paragraph. "We found that the energy bandgap of the PI-derived polymers is not coupled with the dihedral angle between adjacent conjugated planes" Similar conclusions also appear in the abstract and conclusion, as well as on page 9 "which corroborates that by merely focusing on the optimization of E_g may not be able to yield the best high temperature capacitive performance." These conclusions are contrary to machine learning studies that predicted the chemical structure of polymers with high T_g and high E_g . So what is the significance of employing machine learning in this work?
3. There are deficiency and incoherence in how the Supplementary Information has been presented. Many parts in Supplementary Information are disconnected from the main text, calling for additional efforts to improve the coherence and clarity of the whole presentation. For instance, many experimental data in Supplementary Information, such as the dielectric spectrum, dielectric breakdown strength, and mechanical strength of materials are important to this application, but have not been discussed at all in the main text. Furthermore, Section 4 (film quality, which is pretty important) appears

to be a wholly isolated section. Cross-reference to the main text is necessary in order to give the proper context. In addition, the figures in the Supplementary Information are not numbered sequentially, making it unfriendly to the readers. Furthermore,

4. The word “Assembly” in the title is misleading in its current context. It typically correlates with materials with non-covalent connectivity, and thus not appropriate for polymers. A different word of choice is needed to replace “assembly” in the title and throughout the main text.

5. Please don't correlate “conjugation degree” with “dihedral angles between conjugation planes” (at least the way it is presented is confusing). While it is generally true in fully conjugated systems such as organic semiconductors and conjugated polymers, “conjugation” barely applies for any of the polymers discussed here, as there are no actual conjugation between adjacent conjugation planes.

Other comments:

1. On the first page, lines 4-5 of Main, the graceful failure mechanism of polymer dielectrics has been called out. Since the graceful failure feature is important, do the materials reported in this work have this characteristic? It is known that polyimides generally have poor self-healing properties due to the high proportion of carbon in their molecular structure.

2. Page 14, Methods section. The description of machine learning appears to be inadequate for readers. Relevant codes and methods are suggested to be made public. The experimental data of references [11] and [33] are used for machine learning training, however, both references [11] and [33] involved only a very limited set of structures of polyimide or polyetherimide polymers. The authors need to clarify how training is done.

3. The energy bandgap appears in both Figure 2 and Figure 3, are these experimental values or theoretical calculation values? Is there any essential difference between the values obtained by the experimental method and the theoretical calculation method? In addition, in the penultimate line of page 5, “The experimentally determined E_g and T_g are comparable to those predicted by machine learning,” is written. It would be easier if a table or figure is provided for such comparison.

4. Page 7, the last two sentences of the second paragraph are imprecise. Supplementary Fig. S22 and Fig. S23 include 12 different polymers, but Supplementary Table. S1 only has 6 polymers. The authors need to explain the so-called “match”. Additionally, it is recommended to include information for all 12 polymers in Table S1.

5. There are several questions related to the clarity of how the conduction mechanism being determined. Based on Figures 4c, 4d and Supplementary Fig. S24, it is hard to understand why the predominant conduction of polymers with $E_g < 2.9$ eV is Poole-Frenkel emission, and why is hopping conduction the predominant mechanism of polymers with $E_g > 3.2$ eV. How did the authors come to this conclusion? Is it based on fitting results? Why are there both Schottky injection and Poole-Frenkel

emission in Figure 4d? What do these fitted lines represent, and why is there no relevant discussion of Schottky injection in the main text? Also, it feels like related discussion and explanation have been left out in the electric conduction analysis section on page 8. Moreover, are these mentioned conduction mechanisms temperature dependent? The authors only fit the experimental data at 200 degrees. If at different temperatures (room temperature, 150 degrees Celsius or 250 degrees Celsius), is the conclusion still applicable?

6. Page 9, the sentence “It is apparent that the high-temperature energy storage performance (U_e and η) of the polymers with relatively low E_g (< 2.9 eV) complies with the trend of E_g .” is not conclusive. More description and explanation about this trend need to be provided. Furthermore, the properties of these five materials appear to be comparable. Error bars based on multiple experiments are suggested to be provided.

7. Several important references, such as Adv. Mater. 2022, 34, 2207421., where the results with comparable performance are missing for Fig. 5c. While this work reports very high performance, the material does not have a particularly large advantage over the performance reported in the above paper.

8. Statistical electrical breakdown analysis needs to be provided for samples tested at 250°C.

9. SI needs more rigorous revision. For instance, the caption for Supplementary Figure S17a is incorrect. The d-spacing for m-phenyl is actually increasing (decreasing theta), not decreasing! For ssNMR: the chemical shift is plotted backwards — the convention is to increase the chemical shift from right to left. Some peak assignment seems to be hand-waving.

10. Minor grammatic errors. Page 8, line 4, rigidity should be rigid; line 7 from the bottom, more large should be larger.

Reviewer #2 (Remarks to the Author):

Comments on “Tailored assembly of structural units in polymer dielectrics for high-temperature capacitive energy storage”

In this work, the authors have adopted the machine learning (ML) techniques to identify promising polymer structures, which could simultaneously high glass transition temperature T_g and energy band E_g . The polyimide structures have been generated through the combinations of popular structural units based on the diamine and dianhydride molecules. Thus, ML models are used to select about 12 representative polyimides for experimental synthesis and characterization. Some of the polyimides are found to demonstrate unprecedented energy storage performance up to 250 °C. Before this work can be considered for publication, there are several issues to be addressed.

[1] There are a considerable amount of related works published in this area, especially, from Dr. Rampi

Ramprasad's group, such as Sharma, Vinit, Chenchen Wang, Robert G. Lorenzini, Rui Ma, Qiang Zhu, Daniel W. Sinkovits, Ghanshyam Pilania et al. "Rational design of all organic polymer dielectrics." *Nature communications* 5, no. 1 (2014): 1-8; Wu, Chao, Ajinkya A. Deshmukh, Lihua Chen, Rampi Ramprasad, Gregory A. Sotzing, and Yang Cao. "Rational design of all-organic flexible high-temperature polymer dielectrics." *Matter* 5, no. 9 (2022): 2615-2623; Batra, Rohit, Hanjun Dai, Tran Doan Huan, Lihua Chen, Chiho Kim, Will R. Gutekunst, Le Song, and Rampi Ramprasad. "Polymers for extreme conditions designed using syntax-directed variational autoencoders." *Chemistry of Materials* 32, no. 24 (2020): 10489-10500; Alamri, Abdullah, Chao Wu, Ankit Mishra, Lihua Chen, Zongze Li, Ajinkya Deshmukh, Jierui Zhou et al. "Improving the Rotational Freedom of Polyetherimide: Enhancement of the Dielectric Properties of a Commodity High-Temperature Polymer Using a Structural Defect." *Chemistry of Materials* 34, no. 14 (2022): 6553-6558. Just name a few here. It is not clear what is the innovation of the present work, especially, considering that "Papers published by the journal aim to represent important advances of significance to specialists within each field."

[2] The combination of popular structural units (e.g., A1-A11, B2-B10) seems very brutal without clear rationales. How could such a combination ensure the diversity of chemical structures of these polyamides? In particular, a more diverse building block combinations have been explored almost 10 years again (*Nature communications* 5, no. 1 (2014): 1-8).

[3] The authors mentioned that "The Tg polymer dataset with 2989 samples and Eg dataset with 383 samples contain 7 elements, i.e., C, H, O, N, S, F, Cl." Nevertheless, from Fig. 1a-b, there are 4208 data points and 2988 data points for Eg and Tg, respectively. The discrepancy should be explained.

[4] It is not clear why the datasets cover the similar chemical space of these hypothetical polyimides or not. It will be great if the authors could plot the chemical space of the dataset, along with the hypothetical polyimides.

[5] In the recent benchmark study of polymer informatics, the Gaussian processing regression (GPR) model is found not predictive as many other machine learning models (*Journal of Chemical Information and Modeling* 61 (11), 5395-5413. 2021). Thus, the accuracy and uncertainty of the trained machine-learning models should be further discussed.

[6] In Figs. 2 and 3, the authors tried to quantify the impact of different structural units on the Tg and Eg. A more accurate and useful way should be the Explainable AI (XAI), or Interpretable AI technique, such as the SHAP (SHapley Additive exPlanations) analysis.

[7] In Fig. 3b, the authors only compared the synthesized 12 polyamides in the Tg-Eg plot. Nevertheless, there are extensive many other polymers with great Eg and Tg properties. The authors need to add these polymers into this plot as well.

[8] Regarding the MD simulations, Reax force field is not very reliable for these polyimides. The authors should perform the DFT simulations to quantify the energy changes associated with the dihedral angle.

[9] The DSC experiments in the supplementary materials (Fig. S16) seem not very accurate for the Tg estimation. A higher thermal rate (such as 20 or 50 min/°C) might be used to eliminate the noise.

[10] It is not clear whether ML predictions, experimental characterizations, along with MD/DFT simulations are consistent with each other. The authors should further discuss this aspect.

Reviewer #3 (Remarks to the Author):

This paper reports an interesting and novel design strategy of polymer dielectrics for high temperature electrostatic energy storage. A dozen of polyimide polymer structures are selected from 110 candidates by machine learning-driven prediction. Impressively, comprehensive studies of these polymers enable quantitative description of the structural units contributing to the bandgap and glass transition temperature. The most exciting part is that the experimental analysis along with computational simulations reveal that the conduction mechanism is a very important clue to follow, and the dihedral angle between adjacent plane of conjugation can play a key role in determining the high-temperature capacitive performance. This is of substantial difference to the previous understanding, in which bandgap is given the most attention in the design of high-temperature polymer dielectrics for energy storage. The new understanding succeeds in achieving the best energy density and efficiency at high temperature conditions ranging flexibly from 150 to 250 °C through the assembly of structural units. In my opinion, this piece of exciting work would have immediate impact to the community of dielectric energy storage as it offers a completely new thought on the development of high-temperature polymer dielectrics. I recommend to publish this paper in Nat. Commun. There are several minor points that I would suggest the authors to consider or discuss a bit more.

1. Do the best-performed polymers designed from the tailored assembly have adequate mechanical robustness to form capacitor films?
2. How does the dihedral angle between adjacent plane of conjugation affect the stacking of polymer chains? Would that also affect the thermal property of the resultant polymer?
3. I would suggest to add a sentence or two to give an example explaining how the exact value in Fig. 3c is determined.

Reviewer #4 (Remarks to the Author):

This work uses a machine learning approach to predict and screen 12 PI polymers from a library of 110 types of PI-derived polymers, which are then experimentally synthesized and possess a broad range of significantly enhanced electrical and thermal properties, used for high-temperature dielectric capacitive energy storage applications. By combining extensive experimental and theoretical studies, the authors found and established correlations between different structural units of PI and effects on the E_g and T_g of polymers, which are two important factors determining the insulation and thermal stability properties of dielectric polymers. The idea is new, the provided quantitative results that relate structural characteristics with E_g and T_g to some extent are useful for the design and development of high-performance dielectric polymers. The manuscript is well organized and clearly presented. However, the reviewer cannot recommend publications at current version due to some concerns about data collection and interpretation, some revisions should be made. Following are comments for the authors.

1. The bandgap (E_g) is undoubtedly important in this work, however, one concern is that the values of E_g of polymers is not correctly determined by UV-vis absorption characterizations (e.g., Supplementary Figure S15). First, the method to determine optical bandgap used in this work is based on an empirical

formula using the onset of absorbance curves, however, the onset is not clearly or sharply shown in some PI polymers, mistakes can be made here. For example, there is actually a small absorbance increase at ~400 nm in PI-cyc-iso, in addition to a sharp increase at ~300 nm. Tauc plot should be made to obtain accurate E_g values. Or UPS measurements can be conducted to further confirm E_g . Second, the absorbance is pretty high (than 100%) and extremely noisy when wavelength is below 350 nm, indicating unsatisfactory data quality.

2. Why are the computational results of the band gap so different from the UV experimental result? For example, the calculation result of PI is 2.2 eV, and the experimental result is 2.7 eV.

3. Similar problems (to comment 1) can be seen in DSC curves (e.g., Supplementary Figure S16) for determining T_g , which is another important factor in this work. It is very difficult to accurately obtain a T_g in such poor-quality dsc results in some polymer samples, e.g., PI, PI-m, PI-ether. Possible reasons could be sample weight is too low. Besides, by using only DSC data is usually not enough to confidently obtain the real T_g values, DMA measurements could be a second choice as a complementary tool to better determine T_g of polymers.

4. Sometimes molecular weight (M_w) can be critical to electrical and mechanical properties of polymers, yet is not included in this work. The authors are suggested to compared M_w of different PI polymers and see if M_w matters here or can be excluded.

5. The synthesis of PIs involves the selection of different monomers and a complex imidization reaction process. The thermal amination process significantly affects the dielectric properties and energy storage properties of PIs, which has been confirmed by previous work [J. Mater. Chem. A, 2022,10, 10950-10959; Adv. Mater., 2022, 34, 2101976]. In experimental section, all polymers are synthesized by solution method and used the same thermal imidization process, some PIs (e.g., PI (A6-B1), PI-m (A7-B1)) may not be fully imidized because the highest imidization temperature is only 250 °C, which inevitably reduce the accuracy of the experimental results. Is it enough for PI imidization by “a gradient heating method, kept at 100 °C, 150 °C, 200 °C, and 250 °C in sequence for 1 h”? A complete PI imidization is usually obtained by heat treatment to over 300 °C.

6. Supplementary Figure S47 (a), the scale bar is not inconsistent with the caption.

Reviewer #1 (Remarks to the Author):

The authors reported a rational approach to understand and design polyimide polymers for high temperature dielectrics. Based on the machine-learning results, PIs with decent E_g s and T_g s were identified and synthesized. Experimental studies indicated that the conduction mechanism altered from Poole-Frenkel emission to hopping once above an empirical threshold (3.2 eV), and in the hopping region, the dihedral angle between adjacent conjugated ring systems becomes decisive to the trap depth and hence the leakage current density. The findings enabled the design and synthesis of new PI polymers with impressive high-T energy storage properties. Both the approach and the materials properties are state-of-the-art. Despite some technical inconsistencies, the work is quite commendable for the systematic approach, and the main findings could be inspirational for designing aromatic main chain dielectric polymers. I thus recommend its acceptance after the authors address the following comments:

Main concerns:

1. In Figure 5b, the withstand voltage of PI-oxo-iso material is very high, 700 MV/m. However, in Supplementary Fig. S22, the breakdown strength of this material at the same temperature is only 606 MV/m, which is much lower than 700 MV/m. The inconsistency in the results of this experiment deserves further scrutiny. In other words, can each sample reach 650 or 700 MV/m in capacitor tests? This raises the concern about the lack of error analysis, as error bars are missing from the major performance plots presented in Figure 5. Since this work reports many numerical values, it is critical to know if the results are average quantities or best-case values. Although the authors state that 15 specimens were tested for each polymer in the electric breakdown measurements, it is not obvious whether the reported values for other characterizations are representative.

Response: As the reviewer pointed out, the breakdown strength value is lower than the withstand voltage in the $D-E$ loops. This is because of the different process in these two tests.

In fact, in the breakdown strength test (Figure S29), we apply the DC voltage in the rate of 500 V/s. The sample thickness is around 6 microns, so the electrical stress is applied for more than 10 seconds on each sample. Moreover, we judge the voltage at the beginning of burr on the I-V curve as the breakdown point (we think that burr means the sample starts to be unstable).

In the $D-E$ loop test, we use a unipolar wave at the frequency of 100 Hz (Figure 5b), which means the voltage is applied on the sample for only 0.01 s. In this case, a slightly unstable current would have little impact on the test results. Therefore, the breakdown strength in Figure S29 is lower than the withstand voltage of $D-E$ loop in Figure 5b.

We have repeated the $D-E$ loop tests of each sample for 6 times, with very high repeatability and negligible error.

To address this comment, we have added a statement cited as “In the breakdown strength test, we judge the voltage at the beginning of burr on the I-V curve as the

breakdown point.” in the figure caption of Figure S29 of the revised Supplementary Information.

2. Page 7, the first sentence of the second paragraph. “We found that the energy bandgap of the PI-derived polymers is not coupled with the dihedral angle between adjacent conjugated planes” Similar conclusions also appear in the abstract and conclusion, as well as on page 9 “which corroborates that by merely focusing on the optimization of E_g may not be able to yield the best high temperature capacitive performance.” These conclusions are contrary to machine learning studies that predicted the chemical structure of polymers with high T_g and high E_g . So what is the significance of employing machine learning in this work?

Response: As mentioned in the manuscript, the role of machine learning in this work is to assist with the **preliminary** screening of the polymer structure. Although only focusing on the optimization of E_g cannot obtain the best high-temperature capacitance performance, the relatively high E_g is still a crucial factor in material design. For example, as one can see, the polymers with E_g below 3.0 eV (Figure 5a) are apparently inferior to those with E_g above 3.3 eV (Figure 5b). Our conclusion in this study is that when E_g is higher than 3.3 eV, the dihedral angle becomes key to the insulation performance. This finding is on the basis that a series of polymers with high E_g (above 3.3 eV) are available to be investigated, which relies on the fast prediction of machine learning.

Besides, the machine learning also helps to screen out polymer structures with a broad range of electrical and thermal properties, which enables grouped comparisons and quantitative description of the impact of each structural unit on the E_g and T_g of polymers, as shown in Figure 3.

Therefore, the significance of employing machine learning lies in two aspects. 1) To screen out polymer structures with a broad range of electrical and thermal properties for quantitative description of the impact of each structural unit. 2) To screen out plentiful enough polymer structures with high E_g and high T_g , helping us ultimately reveal the decisive role of dihedral angle between adjacent conjugated planes.

To address this comment, we have added a statement cited as “By using machine learning, polymer structures with a broad range of electrical and thermal properties are screened out, which enables grouped comparisons and quantitative description of the impact of each structural unit on the E_g and T_g of polymers. Besides, the prediction by machine learning also offers plentiful enough polymer structures with high E_g and high T_g .” in the Conclusions Section of the revised manuscript.

3. There are deficiency and incoherence in how the Supplementary Information has been presented. Many parts in Supplementary Information are disconnected from the main text, calling for additional efforts to improve the coherence and clarity of the whole presentation. For instance, many experimental data in Supplementary Information, such as the dielectric spectrum, dielectric breakdown strength, and mechanical strength of materials are important to this application, but have not been

discussed at all in the main text. Furthermore, Section 4 (film quality, which is pretty important) appears to be a wholly isolated section. Cross-reference to the main text is necessary in order to give the proper context. In addition, the figures in the Supplementary Information are not numbered sequentially, making it unfriendly to the readers.

Response: We thank the reviewer for pointing out this issue.

To address this comment, the main text and Supplementary Information have been revised, including the description of breakdown strength, dielectric properties and mechanical strength in the main text, the cross-reference with the main text in terms of film quality, and the rearrangement of the figures in the Supplementary Information.

4. The word “Assembly” in the title is misleading in its current context. It typically correlates with materials with non-covalent connectivity, and thus not appropriate for polymers. A different word of choice is needed to replace “assembly” in the title and throughout the main text.

Response: Thanks for this suggestion.

To address this comment, the word “assembly” is replaced with “combination” in the title and throughout the main text of the revised manuscript.

5. Please don't correlate “conjugation degree” with “dihedral angles between conjugation planes” (at least the way it is presented is confusing). While it is generally true in fully conjugated systems such as organic semiconductors and conjugated polymers, “conjugation” barely applies for any of the polymers discussed here, as there are no actual conjugation between adjacent conjugation planes.

Response: We thank the reviewer for pointing out this issue.

To address this comment, we have checked and revised the relevant incorrect expressions in the main text of the revised manuscript.

Other comments:

1. On the first page, lines 4-5 of Main, the graceful failure mechanism of polymer dielectrics has been called out. Since the graceful failure feature is important, do the materials reported in this work have this characteristic? It is known that polyimides generally have poor self-healing properties due to the high proportion of carbon in their molecular structure.

Response: We agree that polyimides generally have poor self-healing properties.

To address this comment, we have carried out the self-healing test of the material. The results show that only 20% of PI-oxo-iso have self-healing properties in the 10 D-E loop tests, and the energy storage performance after self-healing is significantly degraded, which may be due to the high residual carbon content. Therefore, we change the statement in the main text into “Polymer dielectrics have been the materials of choice for high-voltage dielectric capacitors by virtue of their high voltage endurance and ease of processing”.

2. Page 14, Methods section. The description of machine learning appears to be inadequate for readers. Relevant codes and methods are suggested to be made public. The experimental data of references [11] and [33] are used for machine learning training, however, both references [11] and [33] involved only a very limited set of structures of polyimide or polyetherimide polymers. The authors need to clarify how training is done.

Response: According to the reviewer's suggestion, we corrected the description of machine learning method and explained in detail how the PI structure is trained. The code can be easily obtained based on the description. Although references [11] and [33] only contain three polymer molecular structures, we use the K-fold method to make the trained E_g close to experimental values, where all the data except for those of the PI-derived polymers in the training dataset are divided into five parts and the data of the PI-derived polymers are trained in every part. The K-fold method is known to improve the simulation accuracy [ref.28 and 30 in the revised manuscript].

To address this comment, we have added a statement cited as “Due to fact that the calculated E_g values are usually lower than the experimental ones, we replaced some E_g values of PI-derived polymers in the training dataset with experimental data to improve the accuracy of prediction. We also took the K-fold ($k=5$) method, where all the data except for those of the PI-derived polymers in the training dataset are divided into five parts and the data of the PI-derived polymers are trained in every part. The dataset has been plotted in Fig. S1 where the PI-derived polymers are highlighted.” in the Methods Section of the revised manuscript.

Supplementary Figure S1 Chemical space of (a) E_g training set (4208 polymers) and (b) T_g training set (2988 polymers) (gray circles) considered the PI system polymer (highlighted), illustrated using the first two principal components (pc1 and pc2).

3. The energy bandgap appears in both Figure 2 and Figure 3, are these experimental values or theoretical calculation values? Is there any essential difference between the values obtained by the experimental method and the theoretical calculation method? In addition, in the penultimate line of page 5, “The experimentally determined E_g and

T_g are comparable to those predicted by machine learning,” is written. It would be easier if a table or figure is provided for such comparison.

Response: The E_g in Figure 2 is the predicted result of machine learning (calculated result), and the E_g in Figure 3 is obtained from the UV-vis test (experimental result).

To address this comment, we have provided an additional figure (Figure S19) to illustrate the difference between the theoretical calculation value and the experimental result value in the revised Supplementary Information.

Supplementary Figure S19 Comparison of T_g and E_g of the selected 12 polymers between the experimental method and the theoretical calculation method.

4. Page 7, the last two sentences of the second paragraph are imprecise. Supplementary Fig. S22 and Fig. S23 include 12 different polymers, but Supplementary Table. S1 only has 6 polymers. The authors need to explain the so-called "match". Additionally, it is recommended to include information for all 12 polymers in Table S1.

Response: Considering that the dihedral angle may only affect the molecular structure whose conduction mechanism is hopping, we only list the 6 polymers with E_g above 3.3 eV in Table S2.

To address this comment, we have corrected the expression into “Interestingly, it was unveiled that the trends of leakage current density (Supplementary Fig. S25) of the PI-derived polymers with E_g above 3.3 eV match well with that of θ (Supplementary Tab. S2).” in the revised manuscript. Besides, according to the suggestion of the reviewer, we have added all 12 polymers to Table S1 of the revised Supplementary Information.

5. There are several questions related to the clarity of how the conduction mechanism being determined. Based on Figures 4c, 4d and Supplementary Fig. S24, it is hard to understand why the predominant conduction of polymers with $E_g < 2.9$ eV is Poole-Frenkel emission, and why is hopping conduction the predominant mechanism of polymers with $E_g > 3.2$ eV. How did the authors come to this conclusion? Is it based

on fitting results? Why are there both Schottky injection and Poole-Frenkel emission in Figure 4d? What do these fitted lines represent, and why is there no relevant discussion of Schottky injection in the main text? Also, it feels like related discussion and explanation have been left out in the electric conduction analysis section on page 8. Moreover, are these mentioned conduction mechanisms temperature dependent? The authors only fit the experimental data at 200 degrees. If at different temperatures (room temperature, 150 degrees Celsius or 250 degrees Celsius), is the conclusion still applicable?

Response: Firstly, there are mathematical models for Poole-Frenkel emission, Schottky injection and hopping conduction. Schottky injection current density can be expressed by,

$$J = AT^2 \exp\left(\frac{-q\left(\phi_B - \sqrt{qE / 4\pi\epsilon_r\epsilon_0}\right)}{k_B T}\right) \quad (1)$$

$$A = \frac{4\pi q k_B^2 m_0}{h^3} \quad (2)$$

$$\ln\left(\frac{J}{T^2}\right) = \left(\frac{\sqrt{q^3 / 4\pi\epsilon_r\epsilon_0}}{k_B T}\right) \sqrt{E} + \ln(A) - \left(\frac{q\phi_B}{k_B T}\right) \quad (3)$$

Therefore, for Schottky injection, $\ln(J)$ is linear with $E^{1/2}$. The linear fitting results of $\ln(J)$ on $E^{1/2}$ can be used to determine whether the Schottky injection is the dominant conduction mechanism. However, it should be noted that the fitting slope is related to the dielectric constant of the material. Even if the fitting degree is high, if the dielectric constant derived from the slope does not match the experimental dielectric constant, it is not in the mechanism of Schottky injection. For example, in Figure S26b, $\ln(J)$ is linear with $E^{1/2}$, but the dielectric constant calculated by the slope is only about 1, so it does not conform with Schottky injection.

Similarly, Poole-Frenkel emission current density can be given as,

$$J = q\mu N_c E \exp\left(\frac{-q\left(\phi_T - \sqrt{qE / \pi\epsilon_r\epsilon_0}\right)}{k_B T}\right) \quad (4)$$

$$\ln(J/E) = \frac{\sqrt{q^3 / \pi\epsilon_r\epsilon_0}}{k_B T} \sqrt{E} + \ln(q\mu N_c) - \frac{q\phi_T}{k_B T} \quad (5)$$

In which $\ln(J/E)$ is linear with $E^{1/2}$. The linear fitting results of $\ln(J/E)$ on $E^{1/2}$ and the dielectric constant derived from the slope can be used to determine whether the Poole-Frenkel emission is the dominant conduction mechanism. Therefore, by fitting the leakage current-electric field plot, it can be proved that the conduction mechanism of polymers with $E_g < 3.0$ eV changes from Schottky injection to Poole-Frenkel emission with the increase of electric field.

The current density generated by hopping conductance can be expressed as,

$$J = 2nq\lambda\nu \exp\left(-\frac{E_a}{k_B T}\right) \sinh\left(\frac{\lambda qE}{2k_B T}\right) \quad (6)$$

$$J = \alpha \sinh(\beta E) \quad (7)$$

$$\alpha = 2nq\lambda v \exp\left(-\frac{E_a}{k_B T}\right) \quad (8)$$

$$\beta = \frac{\lambda q}{2k_B T} \quad (9)$$

The leakage current density measured under different electric field is fitted to determine whether the dominant conduction mechanism is the hopping conduction. In the formula, the fitting parameter β is a constant related to the hopping distance, which can be used to calculate the hopping distance. As shown in Figure 4d, the hopping conduction the most plausible mechanism for polymers with $E_g > 3.3$ eV.

Generally, with the increase of temperature or electric field, the main conduction mechanism of materials will change. In this case, under low electric field, there are very few free charge carriers in the material, so the leakage current mainly originated from the charge carriers injected from the metal electrode (Schottky injection). However, when the electric field is further increased, a large number of charge carriers used to be trapped in the material obtain enough energy to "detrapped" to the conduction band for transmission, and thus Poole-Frenkel emission will be the dominant conduction mechanism. Therefore, the transition of conduction mechanism depends on whether the carrier can obtain enough energy to transition to the conduction band.

The above mathematical models show that the three kinds of conduction mechanism are temperature dependent. We choose to fit the experimental data at 200 °C for the following reasons: (1) The Schottky injection and Poole-Frenkel emission generally exist in high temperature; (2) The purpose of this work is to design polymer dielectrics for high-temperature operation, so it is more meaningful to understand the conduction mechanism of materials at higher temperatures, because the electron motion is more intense at high temperatures; (3) The leakage current of most of the materials studied in this work cannot be tested at 250 °C, and are extremely vulnerable to breakdown, which is not conducive to understanding the impact of structural units on charge conduction.

To address this comment, we have added the statements cited as “Since the leakage current of the material increases exponentially with the temperature, it is more meaningful to explore the conduction mechanism at higher temperature.”, “To be specific, in the PI-derived polymers with rigid dianhydride (with $E_g < 3.0$ eV), the fitting of the leakage current density versus electric field curve measured under 200 °C shows the transition from Schottky injection to Poole-Frenkel emission with the increase of electric field.” and “These results, along with the fundamental difference between Schottky injection/Poole-Frenkel emission and hopping conduction suggest that, in the case of polymers possessing relatively low $E_g (< 3.0$ eV), electrons can gain enough energy under high temperature and high electric field to be injected from the electrode into the dielectric or to be excited from the valence band to the conduction band, whereas in the case of polymers with relatively high $E_g (> 3.3$ eV), they are more likely to be conducted through the tunneling effect between

adjacent traps.” in the Page 8 of the main text.

6. Page 9, the sentence “It is apparent that the high-temperature energy storage performance (U_e and η) of the polymers with relatively low E_g (< 2.9 eV) complies with the trend of E_g .” is not conclusive. More description and explanation about this trend need to be provided. Furthermore, the properties of these five materials appear to be comparable. Error bars based on multiple experiments are suggested to be provided.

Response: We agree that here the trend of high high-temperature energy storage performance should be made more specific.

To address this comment, we have added error bars in Figure 5, Figure S45 and Figure S66 of the revised manuscript. In addition, for polymers with relatively low E_g (< 3.0 eV), we have added more descriptions cited as “As E_g increases from 2.75 to 2.99 eV, the U_e and η of polymers gradually increase from 0.6 J/cm³ and 25% to 0.9 J/cm³ and 59%, respectively, which can be attributed to dependence of Schottky injection barrier and Poole-Frenkel emission activation energy on the E_g .” to explain that E_g conforms well with the trend of high-temperature energy storage performance.

Figure 5 High-temperature energy storage performance.

Supplementary Figure S45 Field-dependent charge-discharge efficiency and discharged energy density of the various polymer dielectrics at (a) 150 °C with 10 Hz, (b) 150 °C with 100 Hz, (c) 200 °C with 10 Hz.

Supplementary Figure S66 Discharged energy density and charge-discharge efficiency as a function of electric field of PI-oxo-iso and PI-CF₃-iso at 200 °C.

7. Several important references, such as Adv. Mater. 2022, 34, 2207421., where the results with comparable performance are missing for Fig. 5c. While this work reports very high performance, the material does not have a particularly large advantage over the performance reported in the above paper.

Response: We thank the reviewer for providing this information. The unique advantage of our work is to achieve excellent energy storage performance in pure polymers, which is conducive to industrial production and has great potential for further modification.

To address this comment, the reference (Adv. Mater. 2022, 34, 2207421) suggested by the reviewer has been added in Figure 5c for comparison (F-PI/PCBM).

Figure 5c Comparison of the maximum discharged energy density at above 90% efficiency between the PI-oxo-iso and the state-of-the-art polymer-based high-temperature dielectrics at 200 °C with 10 or 100 Hz applied electric field.

8. Statistical electrical breakdown analysis needs to be provided for samples tested at 250 °C.

Response: Thanks for this suggestion.

To address this comment, the electrical breakdown analysis at 150 °C and 250 °C has been added in the revised Supplementary Information as Fig. S62.

Supplementary Figure S62 Weibull distribution analysis of the breakdown strengths of the tailored polymers at (a) 150 °C and (b) 250 °C.

9. SI needs more rigorous revision. For instance, the caption for Supplementary Figure S17a is incorrect. The d-spacing for m-phenyl is actually increasing (decreasing theta), not decreasing. For ssNMR: the chemical shift is plotted backwards — the convention is to increase the chemical shift from right to left. Some peak assignment seems to be hand-waving.

Response: We thank the reviewer for pointing out this issue.

To address this comment, the Supplementary Information have been carefully checked and revised. The caption for Supplementary Figure S20 has been revised, and the chemical shift of ssNMR has also been corrected.

10. Minor grammatical errors. Page 8, line 4, rigidity should be rigid; line 7 from the bottom, more large should be larger.

Response: We thank the reviewer for pointing out this issue. Actually, in page 8, line 7 from the bottom, “more large dihedral angles” means a greater number of large dihedral angles.

To address this comment, we have carefully checked throughout the text, and have corrected the grammatical errors. Besides, “more large dihedral angles” has been corrected as “a greater number of large dihedral angles”.

Reviewer #2 (Remarks to the Author):

Comments on “Tailored assembly of structural units in polymer dielectrics for high-temperature capacitive energy storage”

In this work, the authors have adopted the machine learning (ML) techniques to identify promising polymer structures, which could simultaneously high glass transition temperature T_g and energy band E_g . The polyimide structures have been generated through the combinations of popular structural units based on the diamine and dianhydride molecules. Thus, ML models are used to select about 12 representative polyimides for experimental synthesis and characterization. Some of the polyimides are found to demonstrate unprecedented energy storage performance up to 250 °C. Before this work can be considered for publication, there are several issues to be addressed.

[1] There are a considerable amount of related works published in this area, especially, from Dr. Rampi Ramprasad’s group, such as Sharma, Vinit, Chenchen Wang, Robert G. Lorenzini, Rui Ma, Qiang Zhu, Daniel W. Sinkovits, Ghanshyam Pilania et al. "Rational design of all organic polymer dielectrics." *Nature communications* 5, no. 1 (2014): 1-8; Wu, Chao, Ajinkya A. Deshmukh, Lihua Chen, Rampi Ramprasad, Gregory A. Sotzing, and Yang Cao. "Rational design of all-organic flexible high-temperature polymer dielectrics." *Matter* 5, no. 9 (2022): 2615-2623; Batra, Rohit, Hanjun Dai, Tran Doan Huan, Lihua Chen, Chiho Kim, Will R. Gutekunst, Le Song, and Rampi Ramprasad. "Polymers for extreme conditions designed using syntax-directed variational autoencoders." *Chemistry of Materials* 32, no. 24 (2020): 10489-10500; Alamri, Abdullah, Chao Wu, Ankit Mishra, Lihua Chen, Zongze Li, Ajinkya Deshmukh, Jierui Zhou et al. "Improving the Rotational Freedom of Polyetherimide: Enhancement of the Dielectric Properties of a Commodity High-Temperature Polymer Using a Structural Defect." *Chemistry of Materials* 34, no. 14 (2022): 6553-6558. Just name a few here. It is not clear what is the innovation of the present work, especially, considering that “Papers published by the journal aim to represent important advances of significance to specialists within each field.”

Response: Thank you very much for providing these excellent works. In this manuscript, the role of machine learning is to assist with the preliminary screening of the polymer structure with a wide range of electrical and thermal properties, and the machine learning method itself is not the innovation of this work. The innovation of this work is to put forward a novel systematic approach, by which new scientific understanding is gained and new design principle of material is demonstrated.

To be specific, the novel systematic approach can be briefly described here as three steps, *i.e.*, 1) preliminary screening of the polymer structure using machine learning, 2) in-depth experimental investigation for the quantitative determination of the impact of each structural unit, and 3) tailored combination of the structural units according to the temperature requirement for the design of new polymer. We believe this approach has not been reported before for the design of high-temperature capacitor dielectrics.

The new scientific understanding is, when the bandgap exceeds 3.3 eV, the conduction mechanism of the polymer will change to hopping conduction (low correlation with bandgap), then the dihedral angle of the adjacent conjugate plane will become a more important factor in the high-temperature insulation performance than bandgap, which is fundamentally different from the design concept of previous literature. Based on this new design principle, we designed several pure polymers with record-high energy storage performance at 150 °C, 200 °C and 250 °C, respectively. This discovery also has the potential to be applied to other polymer systems, helping to design molecular structures that can be applied at higher temperatures.

To address this comment, we have cited the relevant works provided by the reviewer at the appropriate site in the Machine Learning Section. Besides, the Conclusions Section has been modified to better reveal the innovation of this work, cited as “A tailored combination of structural units is demonstrated for the design of high-temperature polymer dielectrics. By using machine learning, polymer structures with a broad range of electrical and thermal properties are screened out, which enables grouped comparisons and quantitative description of the impact of each structural unit on the E_g and T_g of polymers. Besides, the prediction by machine learning also offers plentiful enough polymer structures with high E_g and high T_g . In-depth experimental investigation on the electrical properties reveal that the high-temperature insulation performance would experience diminishing marginal utility as the bandgap increases beyond a critical point (about 3.2 eV) in these polymers, where the predominant conduction mechanism changes from Poole-Frenkel emission to hopping. MD and DFT simulations suggest that dihedral angles between adjacent conjugated planes in the molecule play a decisive role in determining the extreme-temperature capacitive performance. This is strikingly different from the prior notion that optimization of E_g would yield the best capacitive performance at high temperature. Based on these understandings, three different polymer structures are designed via the tailored combination of structural units, for operation under 150 °C, 200 °C, and 250 °C, respectively, all exhibiting record discharged energy densities above 90% efficiency under the respective temperature conditions (7.5 J/cm³, 5.3 J/cm³, and 2.1 J/cm³). The strategy reported in this work may offer new possibilities to resolve the challenging dilemma of achieving excellent capacitive performance at the temperature and electric field extremes.”

[2] The combination of popular structural units (e.g., A1-A11, B2-B10) seems very brutal without clear rationales. How could such a combination ensure the diversity of chemical structures of these polyamides? In particular, a more diverse building block combinations have been explored almost 10 years again (Nature communications 5, no. 1 (2014): 1-8).

Response: Firstly, as mentioned in the response to the first comment, the purpose of this study is to demonstrate a novel systematic approach, understand new insights and obtain the high-performance materials. Therefore, the selection of potential polymer structures (the first step in the whole systematic approach) should serve the

purpose of the whole approach. To this end, 110 candidates with E_g ranging from 0.8 eV to 4.2 eV, and T_g ranging from 194 °C to 393 °C were predicted, from which the 12 representative polymers containing 9 structural units were selected to be synthesized. Such diverse of structural unit combinations are satisfactory for the quantitative description of the impact of each structural unit on the E_g and T_g of polymers, which ultimately allows us to gain new understandings and to design new polymers. Since such combinations can well serve the purpose of the whole approach, some uncommon or hardly synthesized monomers are avoided in our selected chemical space of PI-derived polymers for prediction.

Secondly, this study aims at facilitating the large-scale production of high-performance polymer dielectrics for high-temperature energy storage, as mentioned in page 3, paragraph 2 of the manuscript. Therefore, one of the main principles for selecting the structural monomers in this study is that they are commercially available for the convenience of preparation. The monomers are expected to contain as much as possible common structural units (such as ether bond and methylene), similar to the structure of classical PI (Kapton) and PEI (Ultem).

Thirdly, this study is distinctly different from the previous works mentioned by the reviewer. For example, the work reported in Nature communications 5, no. 1 (2014): 1-8 concerns a general computational methodology for the rational design of polymer dielectrics, which involves very few experimental data and no specific capacitive performance is studied. By contrast, our manuscript reports a tailored combination approach of structural units for high-temperature energy storage polymer dielectrics, which mainly relies on the experimental work. In our study, the computational simulation is only employed as a tool that serves the purpose of the whole systematic approach, and the quantitative determination of the impact of each structural unit and tailored combination of the structural units are the more important parts, which are both based on the experimental studies.

To address this comment, we have cited the relevant works provided by the reviewer at the appropriate site in the Machine Learning Section. In addition, we have added a statement cited as “For the convenience of synthesis and large-scale preparation, some uncommon or hardly synthesized monomers are avoided in our selected chemical space of PI-derived polymers for prediction.” in the Page 4 of the revised manuscript.

[3] The authors mentioned that “The T_g polymer dataset with 2989 samples and E_g dataset with 383 samples contain 7 elements, i.e., C, H, O, N, S, F, Cl.” Nevertheless, from Fig. 1a-b, there are 4208 data points and 2988 data points for E_g and T_g , respectively. The discrepancy should be explained.

Response: Thank you very much for your pointing out the problem. We apologize for the incorrect description for the dataset samples.

To address this comment, we have corrected the statement of machine learning method to “The T_g polymer dataset contains 2988 samples collected form PolyInfo dataset, and the E_g polymer dataset contains 4208 samples collected form Kahazana dataset” in the Methods Section of the revised manuscript.

[4] It is not clear why the datasets cover the similar chemical space of these hypothetical polyimides or not. It will be great if the authors could plot the chemical space of the dataset, along with the hypothetical polyimides.

Response: Thanks for this suggestion.

To address this comment, the chemical space of the dataset has been added in the revised Supplementary Information as Fig. S1.

Supplementary Figure S1 Chemical space of (a) E_g training set (4208 polymers) and (b) T_g training set (2988 polymers) (gray circles) considered the PI system polymer (highlighted), illustrated using the first two principal components (pc1 and pc2).

[5] In the recent benchmark study of polymer informatics, the Gaussian processing regression (GPR) model is found not predictive as many other machine learning models (Journal of Chemical Information and Modeling 61 (11), 5395-5413. 2021). Thus, the accuracy and uncertainty of the trained machine-learning models should be further discussed.

Response: We thank the reviewer for pointing out this issue. In the prediction model in this manuscript, neural network prediction model, decision tree model and random forest model are additionally adopted to train the dataset. In the prediction results, the coefficient of determination of the test set is high (> 0.85). However, in different prediction models, the prediction results of 110 sets of T_g and E_g data are biased (~ 20 K for T_g and ~ 0.2 for E_g). Considering the comparison of known experimental results and predicted results, the model Gaussian processing regression (GPR) is adopted in the manuscript.

To address this comment, we have added a statement cited as “Considering that the minor difference between the predicted results and the experimental results is acceptable (Fig. S19), the machine learning method Gaussian processing regression (GPR) model is applied rather than other machine learning methods (neural network, random forest, decision tree, *etc.*)” in the Methods Section of the revised manuscript.

Supplementary Figure S19 Comparison of T_g and E_g of the selected 12 polymers between the experimental method and the theoretical calculation method.

[6] In Figs. 2 and 3, the authors tried to quantify the impact of different structural units on the T_g and E_g . A more accurate and useful way should be the Explainable AI (XAI), or Interpretable AI technique, such as the SHAP (SHapley Additive exPlanations) analysis.

Response: As mentioned in page 5, paragraph 2, the values in Figure 3c are experimental results obtained by ultraviolet-visible (UV-Vis) absorption spectroscopy, differential scanning calorimetry (DSC) and dynamic mechanical analysis (DMA) tests. When the amount of data is large (e.g., 110 polymers), AI technology is a more reliable method. However, in fact, the values in Figure 3c are derived from experimental results (only 12 polymers), not from the predicted results of machine learning. We quantify the impact of structural units on T_g and E_g based on the change of the molecular structure of classical PI/PEI. For example, PI-ether increases the phenyl ether structure compared with PI, PI-methyl replaces the ether bond with the methylene structure compared with PI, and PEI-*p* replaces the *m*-benzene ring with the *p*-benzene ring compared with PEI. Therefore, the values in Figure 3c can be obtained by simply adding and subtracting the T_g and E_g of the 12 polymers.

To address this comment, we have changed “Based on the afore-mentioned results” as “Based on the afore-mentioned experimental results” in the Page 6 of the revised manuscript.

[7] In Fig. 3b, the authors only compared the synthesized 12 polyamides in the T_g - E_g plot. Nevertheless, there are extensive many other polymers with great E_g and T_g properties. The authors need to add these polymers into this plot as well.

Response: Thanks for this suggestion.

To address this comment, we have added some pure polymers (published in the previous literature) to Figure 3b of the revised manuscript.

Figure 3b Comparison of experimental electrical and thermal properties between polymers synthesized in this work and common commercial polymers (*i.e.*, polyethylene glycol terephthalate (PET), polyethylene naphthalate (PEN), poly(ether-ether-ketone) (PEEK), and polyamide-imide (PAI)).

[8] Regarding the MD simulations, Reax force field is not very reliable for these polyimides. The authors should perform the DFT simulations to quantify the energy changes associated with the dihedral angle.

Response: Thanks for this suggestion.

To address this comment, we have replaced the results of energy barrier from the MD simulation with those from the DFT calculation.

Figure 4b The rotational energy barrier of benzene ring as a function of rotation angle.

[9] The DSC experiments in the supplementary materials (Fig. S16) seem not very accurate for the T_g estimation. A higher thermal rate (such as 20 or 50 min/°C) might be used to eliminate the noise.

Response: Thanks for this suggestion. We have tried higher or lower heating rates and found that the T_g of some PI-derived polymers could not be more accurately determined. Therefore, we employ DMA test to help determine the accurate T_g .

To address this comment, the DMA results have been added in the revised Supplementary Information as Fig. S18. In addition, the T_g values in the main text have been revised.

Supplementary Figure S18 Dynamic mechanical analysis (DMA) curves of some polymer dielectrics whose T_g cannot be accurately determined by DSC.

[10] It is not clear whether ML predictions, experimental characterizations, along with MD/DFT simulations are consistent with each other. The authors should further discuss this aspect.

Response: Thanks for this suggestion.

To address this comment, we have provided an additional figure (Fig. S19) in the revised Supplementary Information to illustrate that machine learning and experimental data are consistent. In addition, Table S2 shows that the dihedral angle (calculated by MD/DFT) of materials with bandgap higher than 3.3 eV is also consistent with the trend of leakage current.

Supplementary Figure S19 Comparison of T_g and E_g of the selected 12 polymers between the experimental method and the theoretical calculation method.

Reviewer #3 (Remarks to the Author):

This paper reports an interesting and novel design strategy of polymer dielectrics for high temperature electrostatic energy storage. A dozen of polyimide polymer structures are selected from 110 candidates by machine learning-driven prediction. Impressively, comprehensive studies of these polymers enable quantitative description of the structural units contributing to the bandgap and glass transition temperature. The most exciting part is that the experimental analysis along with computational simulations reveal that the conduction mechanism is a very important clue to follow, and the dihedral angle between adjacent plane of conjugation can play a key role in determining the high-temperature capacitive performance. This is of substantial difference to the previous understanding, in which bandgap is given the most attention in the design of high-temperature polymer dielectrics for energy storage. The new understanding succeeds in achieving the best energy density and efficiency at high temperature conditions ranging flexibly from 150 to 250 °C through the assembly of structural units.

In my opinion, this piece of exciting work would have immediate impact to the community of dielectric energy storage as it offers a completely new thought on the development of high-temperature polymer dielectrics. I recommend to publish this paper in Nat. Commun. There are several minor points that I would suggest the authors to consider or discuss a bit more.

1. Do the best-performed polymers designed from the tailored assembly have adequate mechanical robustness to form capacitor films?

Response: As shown in Fig. S23, the best-performed polymer PI-oxo-iso comparable mechanical properties with PEI, suggesting it has adequate mechanical robustness to form capacitor films. The other two best-performed polymers designed

from the tailored assembly have also been proved to have adequate mechanical robustness to form capacitor films through mechanical property test.

To address this comment, the mechanical properties of PEI-iso and PI-CF₃-iso have been added in the revised Supplementary Information as Fig. S57. In addition, we have added a statement cited as “The result from mechanical test shows that all the high-performance polymers including PI-oxo-iso, PEI-iso and PI-CF₃-iso have comparable mechanical strength to the regular PEI, which has been proved to be able to form thin capacitor films.” in the revised manuscript.

Supplementary Figure S57 Stress strain curve and Young's modulus of PEI-iso and PI-CF₃-iso.

2. How does the dihedral angle between adjacent plane of conjugation affect the stacking of polymer chains? Would that also affect the thermal property of the resultant polymer?

Response: The dihedral angle between adjacent conjugate planes does not affect the stacking of polymer chains. As shown in Figure S17, only *m*-benzene shows increased molecular chain spacing in all structural units, so we believe that the dihedral angle between adjacent conjugate planes will not affect the thermal property of the resultant polymer.

To address this comment, we have added the statement cited as “Except for *m*-benzene structure, other structural units have minor impact on the stacking of molecular chains, indicating that the rigidity of structural units is the main factor affecting the T_g of polymers.” in the figure caption of Figure S20 of the revised Supplementary Information.

3. I would suggest to add a sentence or two to give an example explaining how the exact value in Fig. 3c is determined.

Response: Thanks for this suggestion.

To address this comment, we have added the statement cited as “For instance, the quantitative impact of adding an ether bond on the E_g and T_g values can be obtained by subtracting the data of PI-ether and PI. Similarly, by subtracting the data of PI-iso and that of PI-ether then dividing them by 2, we can get the quantitative impact of isopropyl relative to ether bond on the E_g and T_g values.” in the Page 6 of revised manuscript.

Reviewer #4 (Remarks to the Author):

This work uses a machine learning approach to predict and screen 12 PI polymers from a library of 110 types of PI-derived polymers, which are then experimentally synthesized and possess a broad range of significantly enhanced electrical and thermal properties, used for high-temperature dielectric capacitive energy storage applications. By combining extensive experimental and theoretical studies, the authors found and established correlations between different structural units of PI and effects on the E_g and T_g of polymers, which are two important factors determining the insulation and thermal stability properties of dielectric polymers. The idea is new, the provided quantitative results that relate structural characteristics with E_g and T_g to some extent are useful for the design and development of high-performance dielectric polymers. The manuscript is well organized and clearly presented. However, the reviewer cannot recommend publications at current version due to some concerns about data collection and interpretation, some revisions should be made. Following are comments for the authors.

1. The bandgap (E_g) is undoubtedly important in this work, however, one concern is that the values of E_g of polymers is not correctly determined by UV-vis absorption characterizations (e.g., Supplementary Figure S15). First, the method to determine optical bandgap used in this work is based on an empirical formula using the onset of absorbance curves, however, the onset is not clearly or sharply shown in some PI polymers, mistakes can be made here. For example, there is actually a small absorbance increase at ~400 nm in PI-cyc-iso, in addition to a sharp increase at ~300 nm. Tauc plot should be made to obtain accurate E_g values. Or UPS measurements can be conducted to further confirm E_g .

Response: Thanks for this suggestion.

To address this comment, $(\alpha h\nu)^2-h\nu$ plots by Tauc plot have been added in the revised Supplementary Information as Fig. S16b, and the obtained E_g and T_g are also listed in Table S1.

Second, the absorbance is pretty high (than 100%) and extremely noisy when wavelength is below 350 nm, indicating unsatisfactory data quality.

Response: Thanks for pointing out this issue.

To address this comment, high-quality UV-vis diagram has been added in the revised Supplementary Information as Fig. S16a.

Supplementary Figure S16 (a) UV-vis absorption spectra and (b) $(\alpha h\nu)^2$ - $h\nu$ plots by Tauc plot of the experimental polymer dielectrics in this work, in which α , h , and ν are the absorption coefficient, Planck constant, and light frequency, respectively.

2. Why are the computational results of the bandgap so different from the UV experimental result? For example, the calculation result of PI is 2.2 eV, and the experimental result is 2.7 eV.

Response: As the reviewer pointed out, the calculation result of PI is 2.2 eV, much lower than the experimental result (2.7 eV). This is because of the inherent problem of first-principle calculation.

In the DFT, the resolution of Kohn Sham equation does not consider the excited state of the system, which usually obtains the lower energy level above the valence band, resulting in the underestimation of the bandgap. Although we have adopted the PAW and HSE06 methods for correction, the calculated bandgap is still 20%-30% lower than the experimental results, which also occurs in many literature works [Ref. 30-33]. Therefore, the energy bandgap values calculated by DFT in this work are only used for qualitative analysis, *i.e.*, determining the impact of benzene rotation angle on the energy bandgap.

To address this comment, we have added the statement cited as “However, the bandgap values of polymers are still underestimated. The results are only used for qualitative analysis here.” in the Methods Section of revised manuscript.

3. Similar problems (to comment 1) can be seen in DSC curves (*e.g.*, Supplementary Figure S16) for determining T_g , which is another important factor in this work. It is very difficult to accurately obtain a T_g in such poor-quality dsc results in some polymer samples, *e.g.*, PI, PI-m, PI-ether. Possible reasons could be sample weight is too low. Besides, by using only DSC data is usually not enough to confidently obtain the real T_g values, DMA measurements could be a second choice as an complementary tool to better determine T_g of polymers.

Response: Thanks for this suggestion.

To address this comment, the DMA result has been added in the revised Supplementary Information as Fig. S18. In addition, the T_g values in the main text have been revised.

Supplementary Figure S18 Dynamic mechanical analysis (DMA) curves of some polymer dielectrics whose T_g cannot be accurately determined by DSC.

4. Sometimes molecular weight (M_w) can be critical to electrical and mechanical properties of polymers, yet is not included in this work. The authors are suggested to compared M_w of different PI polymers and see if M_w matters here or can be excluded.

Response: Thanks for this suggestion. the Gel Permeation Chromatography (GPC) test has been performed to determine the molecular weight (M_w). The results show that the M_w of 12 PI-derived polymers are distributed in the range of 200-300 kg/mol, indicating that the impact of this factor on the electrical and mechanical properties of polymers can be excluded.

To address this comment, the result of GPC test has been added in the revised Supplementary Information as Figure S22. In addition, we have added the statement cited as “The results show that the M_w of 12 PI-derived polymers are distributed in the range of 200-300 kg/mol, indicating that the impact of this factor on the electrical and mechanical properties of polymers can be excluded.” in the figure caption.

Supplementary Figure S22 Molecular weight from Gel Permeation Chromatography (GPC) of the experimental polymer dielectrics in this work. The results show that the M_w of 12 PI-derived polymers are distributed in the range of 200-300 kg/mol, indicating that the impact of this factor on the electrical and mechanical properties of polymers can be excluded.

5. The synthesis of PIs involves the selection of different monomers and a complex imidization reaction process. The thermal amination process significantly affects the dielectric properties and energy storage properties of PIs, which has been confirmed by previous work [J. Mater. Chem. A, 2022,10, 10950-10959; Adv. Mater., 2022, 34, 2101976]. In experimental section, all polymers are synthesized by solution method and used the same thermal imidization process, some PIs (e.g., PI (A6-B1), PI-*m* (A7-B1)) may not be fully imidized because the highest imidization temperature is only 250 °C, which inevitably reduce the accuracy of the experimental results. Is it enough for PI imidization by “a gradient heating method, kept at 100 °C, 150 °C, 200 °C, and 250 °C in sequence for 1 h”? A complete PI imidization is usually obtained by heat treatment to over 300 °C.

Response: As the reviewer pointed out, a complete PI imidization is usually obtained by heat treatment to over 300 °C. However, during the experiment, we found that the classical PI/PEI synthesized under our heating process, *i.e.*, keeping at 100 °C, 150 °C, 200 °C, and 250 °C in sequence for 1 h, has the best capacitive performance, as shown in the figure below. Therefore, we carried out completely consistent temperature gradient on all polymers, which also avoids the influence of different thermal history on the results. Moreover, in the reference [Adv. Mater., 2022, 34, 2101976] mentioned by the reviewer, 0.87PI-0.13PAA without 300 °C treatment is better than the pure PI with completely imidization after 300 °C treatment, which is consistent to our experimental result.

6. Supplementary Figure S47 (a), the scale bar is not inconsistent with the caption.

Response: We thank the reviewer for pointing out this issue.

To address this comment, we have corrected the scale bar in Figure S49 (a).

REVIEWERS' COMMENTS

Reviewer #1 (Remarks to the Author):

The authors have done an excellent job in responding to the reviewers' comments. I recommend acceptance as is.

Reviewer #2 (Remarks to the Author):

All the concerns have been well addressed. The reviewer would like to suggest the publication of this work.

Reviewer #3 (Remarks to the Author):

It can be accepted for publication.

Reviewer #4 (Remarks to the Author):

The authors well addressed the reviews's comments. The revised version is satisfied for publication